# n-3 Polyunsaturated Fatty Acids Modulate LPS-Induced ARDS and the Lung–Brain Axis of Communication in Wild-Type versus Fat-1 Mice Genetically Modified for Leukotriene B4 Receptor 1 or Chemerin Receptor 23 Knockout

**DOI:** 10.3390/ijms241713524

**Published:** 2023-08-31

**Authors:** Jessica Hernandez, Julia Schäffer, Christiane Herden, Fabian Johannes Pflieger, Sylvia Reiche, Svenja Körber, Hiromu Kitagawa, Joelle Welter, Susanne Michels, Carsten Culmsee, Jens Bier, Natascha Sommer, Jing X. Kang, Konstantin Mayer, Matthias Hecker, Christoph Rummel

**Affiliations:** 1Institute of Veterinary Physiology and Biochemistry, Justus Liebig University Giessen, 35392 Giessen, Germany; jessica.hernandez-2@vetmed.uni-giessen.de (J.H.); julia.schaeffer@vetmed.uni-giessen.de (J.S.);; 2Excellence Cluster Cardio-Pulmonary Institute (CPI), Universities of Giessen and Marburg Lung Center (UGMLC), Member of the German Center for Lung Research (DZL), Justus Liebig University Giessen, 35392 Giessen, Germanyjens.bier@innere.med.uni-giessen.de (J.B.); natascha.sommer@innere.med.uni-giessen.de (N.S.); 3Institute of Veterinary Pathology, Justus Liebig University Giessen, 35392 Giessen, Germany; christiane.herden@vetmed.uni-giessen.de (C.H.); svenja.koerber@vetmed.uni-giessen.de (S.K.); 4Department of Biomedical Engineering, Osaka Institute of Technology, Omiya, Osaka 535-8585, Japan; 5Institute of Pharmacology and Clinical Pharmacy, Philipps University of Marburg, 35032 Marburg, Germanyculmsee@staff.uni-marburg.de (C.C.); 6Center for Mind Brain and Behavior, Universities Giessen and Marburg, 35032 Marburg, Germany; 7Laboratory for Lipid Medicine and Technology, Department of Medicine, Massachusetts General Hospital and Harvard Medical, Boston, MA 02129, USA; 8Department of Internal Medicine, Justus Liebig University Giessen, 35392 Giessen, Germany; konstantin.mayer@innere.med.uni-giessen.de

**Keywords:** immune-to-brain communication, lung–brain axis, lipopolysaccharide, acute respiratory distress syndrome, cytokines, lipid mediators, omega-3 fatty acids, resolvin E1 receptors

## Abstract

Specialized pro-resolving mediators (SPMs) and especially Resolvin E1 (RvE1) can actively terminate inflammation and promote healing during lung diseases such as acute respiratory distress syndrome (ARDS). Although ARDS primarily affects the lung, many ARDS patients also develop neurocognitive impairments. To investigate the connection between the lung and brain during ARDS and the therapeutic potential of SPMs and its derivatives, *fat-1* mice were crossbred with RvE1 receptor knockout mice. ARDS was induced in these mice by intratracheal application of lipopolysaccharide (LPS, 10 µg). Mice were sacrificed at 0 h, 4 h, 24 h, 72 h, and 120 h post inflammation, and effects on the lung, liver, and brain were assessed by RT-PCR, multiplex, immunohistochemistry, Western blot, and LC-MS/MS. Protein and mRNA analyses of the lung, liver, and hypothalamus revealed LPS-induced lung inflammation increased inflammatory signaling in the hypothalamus despite low signaling in the periphery. Neutrophil recruitment in different brain structures was determined by immunohistochemical staining. Overall, we showed that immune cell trafficking to the brain contributed to immune-to-brain communication during ARDS rather than cytokines. Deficiency in RvE1 receptors and enhanced omega-3 polyunsaturated fatty acid levels (*fat-1* mice) affect lung–brain interaction during ARDS by altering profiles of several inflammatory and lipid mediators and glial activity markers.

## 1. Introduction

The omega (n)-3 polyunsaturated fatty acids (PUFA) α-linolenic acid (ALA), eicosapentaenoic acid (EPA), and docosahexaenoic acid (DHA) are essential PUFAs that have demonstrated several anti-inflammatory effects [1,2,3,4,5,6]. Indeed, diets enriched in n-3 PUFAs have been beneficial in human clinical trials [7]. Derivatives of EPA and DHA such as resolvins (Rv), maresins, and protectins belong to a group of specialized pro-resolving mediators (SPMs), which actively terminate inflammation and promote healing as previously demonstrated in animals and humans [8,9,10,11,12,13,14,15,16,17,18,19,20,21,22,23]. All SPMs have been shown to inhibit neutrophil trans-endothelial migration, enhance macrophage phagocytosis of apoptotic neutrophils [8,9,19], and decrease cytokine production [11,12,13,14,15]. Some derivatives of DHA, such as the D-series of Rv such as RvD1, can reduce the deterioration of tight junctions during inflammation and, therefore, decrease edema in the lungs in a model of lipopolysaccharide (LPS)-induced acute lung injury (ALI) [17]. D-series Rv also play an important role in the development of the brain [18,20] and have been shown to reduce cytokine production in in vitro cultures of microglia cells following treatment with LPS [14]. Moreover, EPA-derived E-series Rv also promote the resolution of inflammation by reducing the migration of immune cells such as neutrophils and enhancing the clearance of apoptotic neutrophils by macrophages, as has been shown in different inflammatory models such as during peritonitis or allergic airway inflammation [8,12,21,22,23,24]. Of the E-series Rv, particular attention has been placed on Resolvin E1 (RvE1), which is derived from EPA [12]. Interestingly, RvE1 has a high potential for the treatment of several lung and brain-inflammatory diseases such as LPS-induced depressive behaviors [25,26].

RvE1 predominantly acts through two receptors, the G-protein coupled receptor chemerin receptor 23 (CR) and the leukotriene B4 (LTB_4_) receptor 1 (LR) [21]. The LR is mainly found on leukocytes and neutrophils but CR is expressed by a variety of other tissues [12,27,28,29,30,31]. Notably, CR is found in several organs, such as the liver, and has, thus far, been identified on many cell types including dendritic cells, monocytes, macrophages, endothelial cells, and to a small degree on neutrophils [12,27,28,29,30,31]. RvE1 has been shown to inhibit tumor necrosis factor (TNF) α-induced activation of the inflammatory nuclear factor (NF) κB pathway through CR while also acting as an antagonist on the LR to dampen pro-inflammatory signals [12,21,32,33,34]. Moreover, Rv are antagonists for the LR and inhibit LTB_4_-induced calcium and NFκB signaling to mediate the resolution of inflammation during murine peritonitis [21,35].

Indeed, previous studies have shown that SPMs such as Rv can be beneficial during neuronal damage by promoting brain cell survival and blocking glial cell cytokine production [14,36,37]. For lung inflammation, SPMs and especially Rv have been well studied during acute respiratory distress syndrome (ARDS) and the restoration of homeostasis [26,38,39,40]. ARDS is a type of acute diffuse lung injury, characterized by inflammation leading to increased pulmonary vascular permeability [41]. Mice expressing the *fat-1* gene and therefore endogenously synthesizing n-3 PUFAs, or mice receiving a fish oil infusion, demonstrated reduced leukocyte invasion, protein leakage, and decreased levels of several pro-inflammatory chemokines and cytokines in the lung tissue during LPS-induced ARDS when compared to solvent-treated or wild-type (WT) ARDS controls [38,39]. Notably, the n-3 PUFA derivative RvE1 increased alveolar fluid clearance and decreased inflammatory signaling in a model for bacterial pneumonia and LPS-induced ARDS [26,38,39,40].

In addition to the lung, other organs such as the kidneys, liver, and even the brain can be affected by ARDS [42,43]. In mice, LPS-induced ARDS can impair the blood–brain barrier (BBB) by reducing tight junction protein expression in endothelial cells, while simultaneously impairing cerebral blood flow [44,45]. Moreover, ARDS patients showed impaired cognitive outcomes due to brain damage assessed by S-100 protein and altered glucose metabolism in the brain [46,47]. Neurocognitive impairments such as compromised memory, attention and concentration, and/or decreased mental processing speed have even been reported in some patients several months or years after recovery from ARDS [48,49]. However, the molecular mechanisms responsible for the lung and brain connection during ARDS, i.e., lung-to-brain signaling, remain to be further clarified.

Since n-3 PUFA and their derivatives have demonstrated anti-inflammatory effects on the brain and given the impact of ARDS on the brain, we were particularly interested to see if n-3 PUFAs reduced inflammation during LPS-induced ARDS. We previously found that genetic n-3 PUFA enrichment accelerates recovery from central nervous system-controlled sickness responses such as fever after i.t LPS installation [39]. Because of its role as a regulator in the resolution of inflammation, particular attention was placed on endogenous RvE1, or more precisely, on its two prominent receptors: CR and LR. The high prevalence of both receptors on immune cells as well as the expression of the CR in the lung ensures their presence during ARDS [50]. In our present experiment, we sought to replicate a model of ARDS using LPS-induced local lung inflammation while assessing the anti-inflammatory roles of n-3 PUFAs with a focus on RvE1 actions through both receptors. To investigate the connection between the lung and brain during ARDS and the therapeutic potential of n-3 PUFAs and its derivatives, genetically n-3 enriched *fat-1* (Fat) versus control (WT) mice with unmodified (Norm) or deficient Rv receptors (CR KO or LR KO mice) were used in this study (Figure 1). Fat mice convert n-6 PUFAs to n-3 PUFAs due to the expression of an n-3 PUFA desaturase [51] and are hence a popular model to investigate n-3 PUFAs. To determine underlying signaling pathways of lung-to-brain communication and its alterations by deficient RvE1 signaling, we assessed the lung inflammatory response by multiplex analyses of lung inflammatory marker proteins. Similarly, the systemic inflammatory response was determined in the liver. In the brain, mRNA expression levels of a large panel of inflammatory marker proteins served to screen the effects of ARDS on the brain and its modulation by lipid mediators. The hypothalamus was chosen as this brain structure contains pivotal hubs in immune-to-brain communication and sickness signaling during inflammatory peripheral insults [52]. These include so-called sensory circumventricular organs (CVO) such as the vascular organ of lamina terminalis (OVLT) with a leaky BBB prone to detecting circulating mediators [53,54]. Changes in brain and lung lipids, brain inflammatory cytokines, brain oxidative stress, genomic NF-IL6 brain cell activation, and neutrophil recruitment to the brain were further determined through LC-MS/MS, protein carbonyl content (oxidative stress), Western blot, or immunofluorescence staining, as previously reported [52,55,56]. 

## 2. Results

Previous studies have already shown the dynamics of LPS-induced lung inflammation [57,58] Therefore, we focused on changes in the inflammatory response by genetic n-3 enrichment, i.e., Fat background compared to WT counterparts. Moreover, we investigated how RvE1 receptor deficiency altered this response, i.e., CR or LR KO mice. 

### 2.1. Protein Analyses of Inflammatory Mediators in the Lung: n-3 PUFAs and RvE1 Receptors Altered LPS-Induced Pro-Inflammatory Cytokines, Anti-Inflammatory IL-10, and Neutrophil Markers at 0 h, 24 h, and 72 h p.i. in CR and LR KO Mice

Lung cytokine and neutrophil chemoattractant levels assessed by multiplex assays were used to screen for inflammatory modulation by n-3 PUFAs and the RvE1 receptors during LPS-induced ARDS at 0 h, 24 h, and 72 h post inflammation (p.i.). Descriptively, inflammation peaked 24 h after LPS installation into the lung. We found that the main effects due to n-3 PUFAs and deficiencies of both RvE1 receptors were already present at basal levels and persisted until the 72 h time point (Figure 2, Appendix A).

At 0 h, LR KO mice genetically enriched in n-3 PUFAs (Fat background) showed reduced expression of most inflammatory mediators compared to their Norm counterpart. This included pro-inflammatory cytokines (IL-17 (*p* = 0.0010), IL-1β (*p* = 0.0366), TNFα (*p* = 0.0266)), neutrophil chemoattractants (GM-CSF (*p* = 0.0481), CXCL5 (*p* = 0.0447)), and anti-inflammatory IL-10 (*p* = 0.0328). Although inflammation seemed highest at 24 h p.i., specific changes were not observed at this time point save for a CR KO-dependent reduction in IL-10 compared to Norm controls on the WT background (*p* = 0.0125). By 72 h, nearly all differences in inflammatory mediators had ceased and the only reductions could be found in CXCL5 (*p* = 0.0463) and IL-10 (*p* = 0.0079) for the genetically enriched in n-3 PUFAs (Fat background) CR KO group compared to their Norm counterpart. Minor impacts of n-3 PUFAs (i.e., Fat background) were observed with reduced protein levels of the neutrophil markers GM-CSF (0 h, Norm and CR KO), CXCL1 (24 h, Norm and CR KO), and CXCL5 (24 h, Norm and LR KO). However, n-3 PUFAs did not significantly affect other inflammatory mediators in Norm mice, save for an increase in IL-17 (*p* = 0.0267) at basal levels, or RvE1 receptor-deficient mice (Appendix A).

Overall, at basal levels, genetic n-3 PUFA enrichment (Fat background) reduced several inflammatory mediators in mice deficient in the LR compared to Norm controls such as IL-17, IL-1β, TNFα, GM-CSF, CXCL5, and IL-10. This effect of the LR did not extend past the 0 h time point and, during inflammation; the deficiency in the CR became more relevant as seen at 72 h p.i. At the late stage of inflammation, a robust effect of the CR emerged where CR deficiency decreased expression of IL-17, IL-1β, TNFα, GM-CSF, CXCL5, and IL-10 compared to Norm controls (Appendix A), although significant differences between groups were limited. Given the low numbers of animals per group and a high degree of variation, differences could not always be detected. However, the most consistent impact of the CR observed was for the anti-inflammatory cytokine IL-10 where a prolonged reduction in the CR was determined at the 24 h and 72 h time points for this important anti-inflammatory cytokine.

### 2.2. Analyses of Inflammatory Marker Proteins in the Liver: n-3 PUFAs and RvE1 Receptors Altered LPS-Induced Pro-Inflammatory Cytokines, Anti-Inflammatory IL-10, and Neutrophil Markers at 24 h p.i. in Norm, CR, and LR KO Mice

As a proxy for systemic inflammatory spillover of LPS-induced ARDS, changes in cytokine and neutrophil chemoattractant protein levels were also analyzed in the liver. LPS-induced alterations in the production of inflammatory mediators were less pronounced and the ability for genetic n-3 PUFA enrichment to reduce inflammatory mediators was remarkably weaker in the liver than in the lung. Significant differences were restricted to 24 h p.i. and, overall, only a small systemic inflammatory response was detected, which was modestly impacted by n-3 PUFAs and RvE1 receptors (Appendix A).

### 2.3. mRNA Analyses of Hypothalamic Inflammatory Marker Proteins: n-3 PUFAs and RvE1 Receptors Altered LPS-Induced Glial Activation, Neutrophil Markers, Pro-Inflammatory IL-1ß, Anti-Inflammatory IL-1ra and IL-10, Signaling Pathways, and the Enzyme Precursor for Prostaglandin Synthesis mPGES in Norm, CR, and LR KO Mice

Having observed only minor changes in inflammatory marker protein levels in the periphery (i.e., lung and liver) due to LPS-induced ARDS and its modulation by n-3 PUFAs enrichment and RvE1 receptor deficiencies, our next aim was to assess markers for lung-to-brain signaling. To this end, mRNA expression for markers of brain oxidative stress, immune cell trafficking, brain inflammatory responses (cytokines and signaling, enzyme of prostaglandin (PG) synthesis), and markers of neuroplasticity and glial activation were assessed in the hypothalamus (Figure 3). The glial activation marker CD68 did not show any significant changes in the first screening attempt and, thus, further analyses were not performed for microglia (Appendix A). Overall, LPS-induced ARDS also increased markers of inflammation in the brain with a peak at 24 h after stimulation (e.g., IκBα, elastase). Genetic n-3 PUFA enrichment reduced some but not all of the analyzed markers. As previously observed in the lung, we found that the main effects due to n-3 PUFAs and deficiencies of both RvE1 receptors were already present at basal levels and persisted until the 72 h time point (Appendix A).

**Analysis of markers for oxidative stress** included catalase (Cat), superoxide dismutase (SOD)1, SOD2, and nitric oxide synthase (NOS)2. Over the course of inflammation, there were significant alterations by n-3 PUFAs and RvE1 receptor deficiencies (Figure 3, Appendix A). In particular, at 72 h p.i., n-3 PUFA enrichment (Fat background) reduced all markers of oxidative stress when comparing the Norm and CR KO groups regardless of RvE1 receptor expression. However, it was only at basal levels that genetic n-3 PUFA enrichment (Fat background) increased the expression of SOD2 in LR KO mice (*p* = 0.0016). During the late stage of inflammation (72 h), n-3 PUFAs also decreased the production of NOS2 in Norm mice (*p* = 0.0120). Additional comparisons were made between Norm and receptor-deficient mice on either the WT or Fat background for each time point. At 24 h p.i., CR KO mice on a WT background had decreased levels of NOS2 (*p* = 0.0133), while at 72 h p.i. expression levels were decreased for both CR KO (*p* = 0.0049) and LR KO mice (*p* = 0.0068). Further analysis of transcription factor A, mitochondrial (TFAM), a marker of mitochondrial biogenesis and antioxidant effects [59,60], also showed moderate n-3 PUFA-dependent reductions between Norm and LR KO (0 h) and Norm and CR KO (24 h, 72 h) regardless of RvE1 receptor expression (Appendix A). 

Together, these results showed a modest reduction in the expression of markers of oxidative stress due to n-3 PUFAs alone at basal levels and again at 72 h p.i., particularly for SOD2, NOS2, and TFAM. Receptor KO-related differences to Norm mice were restricted to NOS2 and here we saw an earlier impact of CR at 24 h p.i. and CR and LR at 72 h p.i.

**Neutrophils** have been highlighted as an important immune-to-brain signaling pathway during systemic inflammation; therefore, we used the neutrophil marker elastase, neutrophil expressed (Elane) [61], as well as the neutrophil-specific chemokine CXCL1 to gain further insights into their recruitment to the hypothalamus (Figure 3) [55]. Indeed, some main effects of the RvE1 receptor deficiencies and n-3 PUFAs were found in the neutrophil markers (Appendix A). The majority of these effects observed on CXCL1 occurred at 24 h p.i. when impacts of the CR, LR, and n-3 PUFA enrichment were all present. Between Norm mice, at 24 h p.i. n-3 PUFA enrichment did increase the expression of CXCL1 (*p* = 0.0248). This n-3 PUFA-dependent increase was absent in both RvE1 receptor-deficient lines. In fact, in comparison to Fat-CR KO mice, the expression of CXCL1 was significantly lower than in Fat-Norm (*p* = 0.0111), but this was not the case for Fat-LR KO mice. Although there was significant variation in Elane expression between samples, genetic n-3 PUFA enrichment did not seem to have an effect, and the only impact observed was at 24 h p.i. for the LR. The results seen in the neutrophil markers indicate n-3 PUFAs influenced the expression of CXCL1 and deficiency of RvE1 receptors decreased the expression of Elane (LR KO) and CXCL1 (CR KO, LR KO).

**Macrophages** have also been found in the brain perivascular space and choroid plexus during neuroinflammation [62]. Using the macrophage marker CD163 (perivascular macrophages) [63,64], a preliminary analysis was also carried out at 24 h and 72 h p.i. (Appendix A). During late-stage inflammation, n-3 PUFA enrichment did increase the expression of CD163 in Norm mice but there were no effects of RvE1 receptors; therefore, further analysis at 0 h p.i. was not performed. 

**Cytokine expression** in the hypothalamus was determined using a panel of pro- and anti-inflammatory cytokines known to be influential during ARDS, namely, anti-inflammatory IL-10, pro-inflammatory IL-1β and its antagonist IL-1ra (Figure 3), as well as pro-inflammatory IL-6 (Appendix A). It is important to note that the expression of IL-10 was difficult to detect and several samples had to be excluded due to ‘undetermined’ expression values, which could explain a failure to identify significant differences. The main effects of n-3 PUFA enrichment and RvE1 receptor deficiencies were restricted to the IL-1 family (Appendix A). Among the IL-1 family, n-3 PUFA enrichment did increase the expression of IL-1β in Norm mice (*p* = 0.0147) at 24 h p.i. but neither CR nor LR KO mice showed the same response and Fat-CR KO mice even presented significantly reduced expression compared to Fat-Norm (*p* = 0.0191). A similar spike was not seen for IL-1ra although, at the basal level, expression was elevated by n-3 PUFA enrichment in CR KO mice (*p* = 0.0221), significantly higher than the Fat-Norm mice (*p* = 0.0307). Preliminary investigation of IL-6 at 24 h and 72 h did not show significant differences between groups and was, therefore, not analyzed at 0 h p.i. (Appendix A).

**Inflammatory signaling pathways** can be activated by cytokines such as IL-1β and IL-6 via several inflammatory transcription factors such as NF-IL6, NFκB, and signal transducer and activator of transcription (STAT)3 [65,66,67]. In addition, lipid mediators can signal via the anti-inflammatory PPARγ pathway, i.e., the peroxisome proliferator-activated receptor-α coactivator (PGC) 1α, which has been shown to increase in the brain during LPS-induced inflammation [68]. Finally, the enzyme microsomal prostaglandin E synthase (mPGES) is yet another important target gene analyzed due to its role in PG synthesis and fever development [69]. We assessed the expression of NF-IL6, NFκB (via IκBα), PGC1α (Figure 3), and STAT3 (via suppressor of cytokine signaling (SOCS)3) as previously shown during preliminary analyses at 24 h and 72 h p.i. (Appendix A) [68]. Moderate changes due to genetic n-3 PUFA enrichment were detected when comparing Norm and CR KO mice for NF-IL6 (decreased) and Norm and LR KO mice for PGC1α (increased) at the basal levels regardless of RvE1 receptor expression (Appendix A). Moreover, a modest increase in SOCS3 expression was revealed during late-stage inflammation in the LR KO mice regardless of n-3 PUFA status (Appendix A). Together, the apparent peak in NFκbiα at 24 h p.i. is indicative of an increased inflammatory response in the brain. In the absence of significant differences between groups at basal levels, n-3 PUFA enrichment still had a modest influence on mRNA expression of NF-IL6 and PGC1α while mPGES remained unchanged.

**Neuroplasticity and astrocytes** were assessed to identify activation and function within the hypothalamus, expression of brain-derived neurotrophic factor (BDNF), astrocytic marker protein glial fibrillary acidic protein (GFAP) (Figure 3), and microglial activation marker CD68 [70,71] (Appendix A) were analyzed. No effects of either genetic n-3 PUFA enrichment or RvE1 receptors on BDNF or CD68 could be detected but GFAP was altered at the basal level by CR (Figure 3 and Appendix A). Specifically, in n-3 PUFA genetically enriched (Fat background) mice the expression of GFAP was increased in Fat-CR KO mice compared to Fat-Norm at 0 h p.i. (*p* = 0.0162) and again at 72 h p.i. (*p* = 0.0480). Moreover, at 72 h p.i. n-3 PUFAs increased the expression of GFAP in CR KO mice (*p* = 0.0146). We did not detect any changes in CD68 mRNA expression at 24 h or 72 h p.i., thus, we omitted such assessments for further analyses (Appendix A). Based on these results, it appears that genetic n-3 PUFA enrichment increases the mRNA expression of GFAP but only in mice deficient in CR while BDNF remains unaffected.

Overall, in the hypothalamus during LPS-induced ARDS, we found that n-3 PUFA enrichment exerted a broader effect on inflammatory markers and while the effects of the RvE1 receptors were not as present they still occurred in most groups. Although some basal effects of the receptors were detected, the strongest impacts of the receptor deficiencies occurred at 24 h p.i., particularly for the CR KO mice, which had reduced expression of NOS2, CXCL1, and IL-1β compared to Norm controls. At the late stage of inflammation (72 h p.i.), minimal effects were still observed and these were predominantly due to n-3 PUFA enrichment effects on markers for oxidative stress and GFAP expression in CR KO mice.

### 2.4. Protein Analyses of Hypothalamic Oxidative Stress Markers and Immunohistochemical Detection of Total Astrocyte Area in the Paraventricular Nucleus: RvE1 Receptor Deficiency Increased Astrocytic Activation at 24 h p.i but Did Not Alter Mediators for Oxidative Stress in Norm or CR KO Mice

With the observed changes in the mRNA expression of markers for oxidative stress and astrocyte activation, mediator levels in the hypothalamus were further analyzed by Western blot for oxidative stress and immunohistochemistry for total astrocyte area. Since we did not detect major changes in oxidative stress for WT-LR KO or Fat-LR KO mice at 72 h p.i. on mRNA levels during preliminary experiments (Figure 3), we decided to focus on samples from Norm and CR KO mice at 0 h and 24 h p.i. However, we did not reveal significant changes in the content of protein carbonylation, protein levels of catalase and mitochondrial SOD2 in response to genetic n-3 PUFA enrichment (Fat background) or RvE1 receptor deficiency (Appendix A).

As with the oxidative stress markers, a comprehensive analysis of the total astrocyte area was performed focusing on the Norm and CR KO mice. The hypothalamic structure known as the paraventricular nucleus (PVN) was selected for analysis and changes in total astrocyte area were shown over time at 0 h, 24 h, and 72 h p.i., but comparisons were only made at 24 h p.i. (Appendix A). Interestingly, at 24 h p.i. a deficiency in CR changed the percentage of GFAP expression in the PVN where the CR KO groups covered a higher percentage of the PVN in comparison to Norm mice regardless of n-3 PUFA status. The evaluation of astrocyte morphology through GFAP staining in the PVN confirms the mRNA effect we first observed in the hypothalamus (Figure 3) that deficiency in CR increases astrocyte activation.

### 2.5. Immunohistochemical Detection of Inflammatory Transcription Factor Expression at the OVLT, a Brain Structure with a Leaky BBB, and Bifurcation, a Brain Structure with a Complete BBB: Preliminary Results Indicate That n-3 PUFAs and RvE1 Receptors May Alter LPS-Induced NF-IL6 Immunoreactivity at the BIF in Norm and CR KO Mice

To further assess genomic brain cell activation as a marker of humoral immune-to-brain signaling [52], immunohistochemical detection of nuclear NF-IL6 signals was used. Since we did not have enough animals in our WT-Norm control group or the Fat-CR KO group for all time points, the analyses served as preliminary data and were carried out at the level of the OVLT, which is a brain structure with a leaky BBB known to detect circulating mediators [72]. However, semi-quantitative evaluations of NF-IL6 immunoreactivity in the OVLT did not reveal differences between time points (Appendix A) or in groups (Appendix A) possibly related to very low levels of genomic activation per section. Overall, we did not detect significant changes in NF-IL6 activation in the OVLT by n-3 PUFA enrichment or the RvE1 receptor deficiency. Similar to the very modest systemic inflammatory response detected in the liver, these results suggest that circulating cytokines may not play a major role in immune-to-brain communication at the level of the OVLT in the ARDS model applied here.

In addition to the OVLT, a brain structure with a complete BBB was also analyzed, namely the bifurcation (BIF), for differences in NF-IL6 (Figure 4 and Appendix A). A semi-quantitative evaluation was used to assess immunoreactivity levels of NF-IL6 at the BIF. In contrast to the OVLT, at 4 h p.i. NF-IL6 immunoreactivity seemed to peak in WT-Norm and Fat-Norm mice. RvE1 receptor deficiency may affect this process in the CR KO mice since they did not appear to display the same trend in NF-IL6 immunoreactivity, however, in the LR KO mice, there appeared a similar progression in NF-IL6 as seen in the Norm controls. Given that n-3 PUFA enrichment caused modest reductions of NF-IL6 mRNA expression in CR KO mice (Figure 3), immunofluorescence showed that CR but not genetic n-3 PUFA enrichment altered the immunoreactivity at 24 h p.i. (*p* = 0.0197) in the BIF (Figure 5). This effect could not be observed in LR KO mice, where RvE1 receptor-dependent alterations were only significant at 0 h p.i. (*p* = 0.0098) (Appendix A).

Overall, these preliminary descriptive data suggest a modest increase in NF-IL6 activation at 4 h after the lung insult for the BIF, suggestive of some moderate but low systemic inflammatory response (Figure 4) leading to a few humoral signals to the brain in the current model of LPS-induced ARDS. 

### 2.6. Analyses of Immune Cell Recruitment to the OVLT and Bifurcation: Preliminary Results Indicate That LPS Stimulated Neutrophil Recruitment in CR KO Mice May Be Altered in the OVLT by n-3 PUFAs at 4 h p.i. and by CR at 0 h and 120 h p.i. in the Bifurcation. LPS Stimulated Neutrophil Recruitment in CR KO Mice Were Altered by n-3 PUFAs at 0 h and 72 h p.i. and by CR at 0 h, 24 h and 72 h p.i.

To further investigate data on hypothalamic mRNA expression for markers of neutrophil recruitment to the brain during LPS-induced ARDS (Figure 3), we performed immunohistochemical myeloperoxidase (MPO) staining for neutrophils in the OVLT (Figure 6). During early systemic inflammation, such staining reflects neutrophil recruitment to the brain [55]. Once again, the low ‘n’ for our WT-Norm control group and Fat-CR KO group meant that these analyses served to gather preliminary descriptive data. Here, the neutrophil trafficking into the brain appeared to peak at 4 h p.i. in WT-Norm controls but was decidedly absent in the Fat-Norm mice. While mice in the LR KO groups closely mirrored the response of WT-Norm controls, the CR KO groups appeared to have the opposite response in comparison. In the CR KO groups the peak seemed to be present in the n-3 PUFA-enriched (Fat background) CR KO mice while the WT-CR KO mice maintained relatively constant neutrophil levels.

Although neutrophil trafficking to the OVLT occurred at low levels, their recruitment may peak 4 h after LPS stimulation. This response was abolished in Fat-Norm mice and partially recovered with RvE1 receptor deficiency. Significant differences were observed between Norm and CR KO at 0 h p.i. (*p* = 0.0340) (Appendix A) but not LR KO (Appendix A). Alterations between Norm and CR KO mice suggest a role for RvE1 receptors in neutrophil recruitment to the OVLT.

For the BIF, neutrophil recruitment to the brain also appeared to peak at 4 h p.i. in WT-Norm controls while the other groups showed consistently low levels of neutrophils or higher variation per time point (Figure 7). Again, this response was abolished in Fat-Norm mice and partially recovered with RvE1 receptor deficiency. Indeed, the 4 h peaks observed in WT-Norm and Fat-LR KO were absent in Fat-Norm and WT-LR KO mice over the course of the experiment.

Upon further analysis at the level of the BIF, n-3 PUFAs did not significantly affect neutrophil recruitment over time; however, at 0 h (*p* = 0.0441) and 72 h p.i. (*p* = 0.0274) n-3 PUFA enrichment (Fat background) in CR KO mice did show higher levels of neutrophils compared to Fat-Norm controls (Figure 8). Moreover, CR KO mice had an altered response in neutrophil recruitment compared to Norm at 0 h (*p* = 0.0025), 24 h (*p* = 0.0339), and 72 h (*p* = 0.0216). Interestingly, these results were not evident in LR KO mice, which showed no significant impact of n-3 PUFAs or RvE1 receptors (Appendix A).

Similar to the OVLT, n-3 PUFAs alone may inhibit the ability of neutrophils to be recruited to the BIF during ARDS. However, the combination of LR KO and n-3 PUFA expression may enhance recruitment since mice deficient in the LR appeared to have a stronger peak in the n-3 PUFA enriched group. Overall, the observed impact of the CR on neutrophil recruitment to the BIF suggests that modulation of neutrophil recruitment by Rv may be mediated via CR and not LR.

### 2.7. Analyses of Lipid Mediators in the Lung and Brain: n-3 PUFAs and RvE1 Receptors Altered Lipid Mediators at 0 h, 24 h, and 72 h p.i. in CR and LR KO Mice

The lipid mediators in the lung and the brain were analyzed with LC-MS/MS to uncover their potential contributions to observed effects on inflammatory signaling and lung-to-brain communication pathways. Comparisons were made between Norm and receptor-deficient mice on either the WT or Fat background for each time point.

**In the lung**, n-3 PUFAs and RvE1 receptors had an impact on the levels of lipid mediators in the Norm and KO groups (Appendix A). These main effects were present at 0 h, 24 h, and 72 h p.i. due to LPS-induced ARDS. In general, genetic n-3 PUFA enrichment (Fat background) increased levels of lipid mediators during all three time points (Figure 9A). Interestingly, deficiency in RvE1 receptors had strong impacts increasing LTB_4_ and EPA levels, suggesting potential negative feedback mechanisms. 

**Levels of the n-6 PUFA AA derivative LTB_4_**, which binds to the LR, were elevated in CR KO compared to Norm mice (*p* ≤ 0.0001) at basal levels. LTB_4_ decreased in mice with genetic n-3 PUFA enrichment and CR receptor deficiency (*p* ≤ 0.0001). After 24 h, n-3 PUFA enrichment decreased LTB_4_ expression in the LR KO groups (*p* ≤ 0.0001) but, in comparison to WT-Norm mice, levels were elevated in WT-LR KO mice (*p* ≤ 0.0001) (Figure 9A). At 72 h p.i., the levels of LTB_4_ were greatly increased in WT-LR KO (*p* = 0.0073) and Fat-Norm (*p* = 0.0001) compared to WT-Norm mice. Genetic n-3 PUFA enrichment in CR KO and LR KO mice led to a decrease in LTB_4_ (CR KO (*p* = 0.0016) and LR KO (*p* = 0.0104)) (Figure 9A).

**The n-3 PUFA EPA levels** increased significantly with genetic n-3 PUFA enrichment in LR KO in the lung at basal levels (*p* = 0.0024). At 24 h p.i., EPA was increased in mice with genetic n-3 PUFA enrichment and CR deficiency (*p* ≤ 0.0001) and in Norm mice compared to CR KO (*p* = 0.0013) and LR KO (*p* = 0.0008). On the contrary, 72 h p.i. EPA was decreased in LR KO mice with and without genetic n-3 enrichment compared to Norm mice (Fat-Norm vs. Fat-LR KO, *p* = 0.0003; WT-Norm vs. WT-LR KO, *p* = 0.0003) (Figure 9A).

**The EPA derivatives 18-HEPE and RvE1** showed very similar changes in the lung during all time points with almost no differences between the groups at basal levels. At 24 h p.i., 18-hydroxy eicosapentaenoic acid (HEPE) and RvE1 levels increased in Norm groups (18-HEPE, *p* = 0.0451; RvE1, *p* ≤ 0.0001) with genetic n-3 PUFA enrichment compared to mice on the WT background. These elevations of n-3 PUFA EPA derivatives in Norm mice remained until 72 h p.i. (both, *p* ≤ 0.0001). In n-3 genetically enriched CR KO mice, on the one hand, 18-HEPE levels remained mostly the same. RvE1, on the other hand, increased at 24 h p.i. in Fat-CR KO compared to WT-CR KO (*p* ≤ 0.0001) and went back to baseline levels by 72 h p.i (Figure 9A).

**Significant differences in lipid mediators in the brain** were revealed between the groups at all time points (Figure 9B). Both genetic n-3 PUFA enrichment and RvE1 receptor deficiencies had an impact on the levels of lipid mediators in the Norm and KO groups (Appendix A). Due to LPS-induced ARDS, more main effects were present at 24 h and 72 h p.i. In most groups, the amount of lipid mediators increased after the LPS challenge. In general, n-3 PUFAs impacted the levels of lipid mediators to a higher extent than RvE1 receptors. Overall, genetic n-3 PUFA enrichment caused higher levels of lipid mediators during all three investigated time points in Norm and CR KO mice. Interestingly, these effects were reversed in LR KO mice. With the genetic enrichment of n-3 PUFAs in LR KO groups, the levels of lipid mediators were, for the most part, decreased. However, two exceptions were observed in LR KO mice with genetic n-3 enrichment; EPA was increased at 24 h (*p* ≤ 0.0001) and 72 h p.i. (*p* ≤ 0.0001) and 18-HEPE at 0 h p.i. (*p* = 0.0023) (Figure 9B).

**Interestingly the amount of n-6 PUFA AA and its derivative LTB_4_** were higher in Norm mice with genetic n-3 enrichment at 0 h p.i. (AA, *p* ≤ 0.0001; LTB_4_, *p* = 0.0002). Over time, the effect changed such that at 24 h and 72 h p.i. levels increased in Norm and decreased in Fat mice. In response to LPS in WT-LR KO mice, the levels of AA were increased compared to WT-Norm mice at 24 h and 72 h p.i. (24 h, *p* = 0.0004; 72 h *p* = 0.0489) and the levels of LTB_4_ decreased with genetic n-3 enrichment 24 h p.i (*p* = 0.0006) (Figure 9B).

**The n-3 PUFA EPA** is increased in genetic n-3 enriched Norm, CR KO, and LR KO mice at different time points. Genetic n-3 enriched Norm mice demonstrate higher levels of EPA during all three time points (0 h, 24 h, *p* ≤ 0.0001; 72 h, *p* = 0.0211). On the one hand, genetic n-3 enrichment in CR KO mice caused an increase in EPA at basal levels (*p* ≤ 0.0001), on the other hand, in LR KO mice it led to an increase at 24 h (*p* ≤ 0.0001) and 72 h p.i. (*p* ≤ 0.0001). **Its derivative, 18-HEPE,** was increased in genetic n-3-enriched Norm and CR KO mice at 24 h (both, *p* ≤ 0.0001) and 72 h p.i. (Norm, *p* ≤ 0.0001; CR KO, *p* = 0.0023). As previously mentioned, the 18-HEPE derivative can be further converted to RvE1; however, levels of RvE1 were not detectable in any brain sample (Figure 9B).

**The levels of DHA** remained mostly unchanged over time. **The DHA derivative 17(S)-hydroxy docosahexaenoic acid (17(S)-HDHA)** was increased in Norm (*p* = 0.0028) and CR KO (*p* ≤ 0.0001) mice with genetic n-3 enrichment at 0 h p.i. Interestingly, 17(S)-HDHA increased in WT-LR KO mice at 24 h p.i. compared to Fat-LR KO and WT-Norm mice (both, *p* ≤ 0.0001) (Figure 9B). 

The 17(S)-HDHA derivative can be further converted to **neuroprotectin D1 (NPD1) and protectin DX (PDX)** (both mediators cannot be distinguished by LC-MS/MS). Overall, these protectins were increased in Norm and CR KO mice at basal levels with genetic n-3 PUFA enrichment (Norm, *p* ≤ 0.0001; CR KO, *p* = 0.0003), 24 h p.i. (Norm, *p* ≤ 0.0001; CR KO, *p* = 0.0078) and also at 72 h p.i. for Norm mice alone (*p* = 0.0025). Moreover, 17(S)-HDHA can be converted to **RvD1 and RvD2** but only RvD2 was found to be increased in Fat-Norm mice at 0h p.i. (Figure 9B). 

Interestingly, the levels of **14(S)-hydroxy docosahexaenoic acid (14(S)-HDHA)**, another derivative of DHA, were changed in a similar way to 17(S)-HDHA derivatives NPD1 and PDX. The derivate 14(S)-HDHA can be further converted to **Maresin 1** (Mar1) [9]. As such, Mar 1 showed similar alterations as NPD1 and PDX as well. In general, it was increased in most groups with n-3 enrichment at 24 h p.i. but decreased again at 72 h p.i. It is nonetheless interesting to note that genetic n-3 enrichment in CR KO mice led to a very high increase of Mar 1 at 24 h p.i. (*p* ≤ 0.0001) (Figure 9B).

Overall, our data show that EPA and its derivatives are more abundant in the lung than in the brain [73,74] whereas DHA and its derivatives are found at higher levels in the brain (Figure 9), confirming previous reports by others [75,76,77].

## 3. Discussion

In the present study, we are the first to report underlying mechanisms of lung-to-brain signaling during LPS-induced ARDS with respect to the role of cytokines, lipid mediators, and trafficking of neutrophils to the brain. Indeed, LPS-induced lung inflammation did induce signs of brain inflammatory signaling such as NFκB. Overall, hallmarks of low-grade inflammation and other surrogate parameters such as genomic NF-IL6 activation in brain structures with and without complete BBB (i.e., BIF and OVLT) were not detected at levels to allow for a major impact in the lung-to-brain axis. Interestingly, neutrophil trafficking may contribute to underlying mechanisms of lung–brain communication. While descriptive, our data show an LPS-induced increase in neutrophil recruitment at 4 h p.i. at the level of the BIF, which was abolished by genetic n-3 PUFA enrichment in a CR-dependent manner. Moreover, we revealed that during LPS-induced ARDS genetic n-3 PUFA enrichment and the absence of RvE1 receptors, mediators of inflammation in the lung, liver, and hypothalamus were altered. Overall, pro-inflammatory cytokines such as IL-17, IL-1β, and TNFα, as well as the anti-inflammatory cytokine IL-10, were reduced by n-3 PUFAs in RvE1 receptor-deficient mice in the periphery. Additionally, the neutrophil markers GM-CSF and CXCL5 were similarly impacted. Within the brain, TFAM, NF-IL6, and PGC1α were seen to be altered by genetic n-3 PUFA enrichment while CXCL1, IL-1β, IL-1ra, and GFAP were affected by both n-3 PUFA enrichment and the deficient RvE1 receptors. 

Specifically, deficiency of the LR had basal effects in the lung but deficiency of CR showed more significant effects than the LR on pro-inflammatory cytokines and neutrophil markers in the lung during later stages of inflammation. In the brain, impacts of the CR were especially evident for the BIF, where NF-IL6 immunoreactivity and neutrophil recruitment were both altered in mice lacking CR and, in the case of neutrophils, elevated recruitment by n-3 PUFAs was observed in CR but not LR-deficient mice. Moreover, our new insights into differences in lung and brain lipid mediators provide evidence for potential candidates such as 18-HEPE, protectins, RvD2, and Mar1 to explain n-3 PUFA-dependent alterations in lung–brain communication.

During lung inflammation, a consequence of the enhanced leukocyte accumulation and activation is high levels of cytokine synthesis [78,79,80,81]. Notably, IL-1β is one of the key mediators of the lung inflammatory process [81]. A bidirectional interaction exists between the lung and the brain, where insults or injuries to either organ can have impacts on the other [79,80,82,83]. One study performed by Sahu and colleagues (2018) found that following a “two-hit” model for ALI, where mice received an i.t. instillation of hydrochloric acid and LPS, the animals experienced cognitive impairments and elevated levels of systemic pro-inflammatory cytokines [80]. By transmitting information via cytokines and immune cell trafficking, a circuit of communication may be formed between the lung and the brain as has been previously shown with neutrophils and neutrophilic NADPH oxidase, for example, in a mouse model of LPS-induced ALI [82,84,85]. Here, we show only some minor contributions of low-grade systemic inflammation as evidenced in the liver and genomic NF-IL6 brain cell activation as a marker of humoral immune-to-brain communication. 

During ARDS, the release of inflammatory mediators from the lung can influence levels of circulating mediators in the periphery and even reach the brain [84]. By investigating a range of cytokines, we hoped to reveal effects of the RvE1 receptors and/or n-3 PUFA enrichment on lung–brain communication pathways. To assess lung–brain interactions, we first analyzed the overall inflammatory response in the primary site of LPS-induced inflammation in the lung before analyzing the liver, which would provide a systemic cytokine profile. Previous studies have already identified upregulation of the pro-inflammatory cytokines IL-1β, TNFα, and IL-6 in bronchoalveolar lavage fluid (BALF) in ARDS patients [86]. These pro-inflammatory cytokines are able to cross the BBB, especially when permeability increases such as during inflammation [87,88,89]. 

In our present study investigating organ-specific inflammatory responses, nearly all effects were localized to the lung tissue. Fat mice deficient in the LR showed reduced basal levels of pro-inflammatory IL-1β and TNFα along with a reduction in the anti-inflammatory IL-10. Over the course of the experiment, mice deficient in the CR had a consistent reduction in IL-10 suggesting a complex regulation of the pro- and anti-inflammatory balance. On the one hand, even though RvE1 acts as an antagonist for the LR against LTB_4,_ the LTB_4_-BLT1 complex can still signal and exacerbate inflammation as has, for example, been observed in models of myocardial infarction [90]. On the other hand, in addition to RvE1, chemerin can also bind the CR to encourage viral clearance and moderate levels of inflammation to the extent that CR KO mice infected with the mouse pneumonia virus had higher mortality rates than the immune-competent controls [50]. Therefore, in addition to the benefits of RvE1s interaction, the pro- and anti-inflammatory capacities of the LR and CR, respectively, likely contribute to the significance of each KO. In the absence of Rv receptors, however, with the lack of anti-inflammatory Rv signaling, we would have expected a dampened anti-inflammatory response, as previously shown by Arita (2007) in a mouse model of peritonitis and an enhanced inflammatory response, as shown by Bondue (2011) in the mouse model of pneumonia [21,50]. Indeed, reduced lung IL-10 levels support the notion of dampened anti-inflammatory signaling in mice with defective Rv signaling in our present study [91]. Yet, reduced lung IL-1β and TNFα levels seemingly do not fit such simple concepts of reduced anti-inflammatory signaling. Such discrepancies may be linked to the complex dynamics of the inflammatory response and could reflect enhanced inflammation at other time points. We would postulate that at basal levels the LR appears to exert more influence on inflammation as a receptor with pro-inflammatory capacities than the CR while, during later stages of inflammation, the CR continues to significantly impact IL-10 production. However, since the reductions in IL-1β and TNFα were only observed in the Fat-LR KO mice, it is unclear to what extent the receptor plays a role in this process as no effect was observed in Norm mice genetically enriched with n-3 PUFAs (Fat background).

Interestingly, SPMs, namely the E series Rv, have also been shown to reduce levels of IL-1β in lung tissue during the combination of ALI and bacterial pneumonia in part by reducing neutrophil accumulation [40]. Moreover, the regulation of inflammation through TNFα and NFκB prevented transgenic *fat-1* mice from developing streptozocin-induced diabetes during which elevated levels of the RvE1 precursor 18-HEPE were detected [92]. As activated neutrophils are key producers of IL-1β and TNFα during ARDS [93,94], it is possible that RvE1 reduced levels of neutrophils in the lung, leading to a reduction in pro-inflammatory cytokines reported in our present study. RvE1-dependent generation of reactive oxygen species (ROS) and depression of anti-apoptotic signals via LRs increases the apoptosis and clearance of neutrophils during ALI [95]. Moreover, induced phagocytosis of apoptotic neutrophils by macrophages will also increase the production of pro-resolving anti-inflammatory mediators such as IL-10 [95]. Herová and colleagues (2015) found that treating human primary macrophages that expressed the CR in vitro with RvE1 (10 nM) increased IL-10 production [96]. Altogether, the previously described therapeutic effects of RvE1 by neutrophil regulation during ARDS fit well with what we have seen in the lungs of mice in the present study.

Heavy recruitment and infiltration of neutrophils to the lung play a major role in pathogenesis during ARDS and can be monitored using markers associated with neutrophil recruitment and activation [97,98,99,100,101,102,103,104]. IL-17 and GM-CSF in particular have been identified as key modulators of inflammation in the lung whose production can be inhibited by n-3 PUFAs [105]. Consistent with the previous studies, we also saw a reduction in both IL-17 and GM-CSF in the lungs of mice, dependent on genetic n-3 PUFA enrichment. However, although the effects of n-3 PUFAs are apparent, it seems to only be the case in combination with the absence of the LR. As we hypothesized for IL-1β, TNFα, and IL-10, this could also be due to more complex regulatory mechanisms involving additional signaling pathways and binding partners aside from RvE1. Nevertheless, other researchers have found that during allergic airway inflammation, Fat mice have a lower BALF concentration of pro-inflammatory cytokines, including CXCL1 and CCL5, and elevated levels of both RvE1 and PD1 in the lung tissue [106]. Research by Haworth et al. (2008) also identified that treatment with RvE1 could decrease levels of IL-17 in BALF as well as suppress IL-17-producing T helper cells during allergic airway inflammation [11]. It has even been shown that n-3 PUFAs are able to modulate the production of CXCL1 and influence infection outcomes during the early stages of lung infection [107]. However, our present results only show a moderate impact on CXCL1 during the later time point of lung inflammation.

In comparison to the lung, there were far fewer alterations to neutrophil marker expression in the liver. Here, we did still observe the n-3 PUFA-dependent reduction in IL-17 within the Norm group with intact Rv receptors consistent with what others have reported [108,109,110]. The differences between results in the lung and liver may again represent a delay in the inflammatory dynamics with an origin in the lung and some spillover to the circulation inducing a response in the liver. However, the lack of a strong effect in the liver served as an indicator of the systemic inflammatory response suggesting weak contributions of humoral mediators to lung–organ interactions in our model of ARDS.

Towards the later time points, i.e., 72 h p.i., a deficiency in the CR was mainly impacting pro-inflammatory cytokines in the lung including IL-17, IL-1β, and TNFα but also CXCL5 and GM-CSF. Since CR is located on a greater variety of cell types than the LR [12,27,28,29,31,111], it is possible that its impact lasts beyond the initial phases of inflammation once the inflammatory leukocytes have dissipated. Overall, by knocking out the RvE1 receptors in combination with genetic n-3 PUFA enrichment, the mice seemed to experience a dampened immune response. Interestingly, such differences were largely absent in WT and Fat mice with intact Rv receptors. The fact that differences were not observed in mice with the receptors may be due to several factors. In addition to the actions of LTB_4_ and chemerin, the low dose of LPS or the time points selected for analyses could also contribute [50,90]. 

To delineate lung-to-brain communication during ARDS and the ability to cause inflammation, further analyses were performed on the brain. Indeed, we found evidence that LPS-induced ARDS was sufficient to alter mediators of inflammation in the hypothalamus as well as neutrophil trafficking to different brain structures. Key differences were found for markers of oxidative stress, mitochondrial activity, neutrophils, cytokines, and astrocyte activation while little to no changes were seen in signaling pathways, PG synthesis, and neuroplasticity. Additionally, neutrophil trafficking and NF-IL6 immunoreactivity were assessed at the OVLT, with a leaky BBB prone to detecting circulating inflammatory mediators [53,112], and the BIF with a complete BBB. By comparing these two structures, it was possible to evaluate blood-borne signaling in lung-to-brain communication.

Oxidative stress is known to play an important role in the inflammatory response during LPS-induced inflammation in the brain [113,114,115,116]. The extent to which oxidative stress occurs can be calculated by measuring antioxidants and microglial expression of NOS2 [117,118]. While enzymes are useful when carrying out certain physiological functions, exorbitant production of NOS2 or insufficient regulation by antioxidants results in oxidative stress and cellular damage [119,120,121]. The anti-inflammatory effects of RvE1 combined with the presence of CR on glial cells have encouraged investigations into its applications in neuroinflammation [20,47,122,123]. Indeed, Zhang and colleagues (2022) found that, in the brain, RvE1 signaling through the CR ameliorated oxidative stress in a diabetic mouse model [124]. Our own initial evaluation of the hypothalamus showed an overall impact of CR and reduced production of NOS2 in WT-CR KO mice during later time points of LPS-induced ARDS. Moreover, CR also altered the expression of TFAM, which plays a key role in mitochondrial biogenesis and maintenance along with additional antioxidant effects [125,126,127]. To confirm our results on hypothalamic NOS2 mRNA expression, additional analyses on protein level were added. However, neither the carbonylation of proteins nor protein levels of Cat and SOD2 were altered by n-3 PUFA enrichment or CR receptor deficiency. Potentially, discrepancies between mRNA and protein data may be explained due to cell-specific regulations that were diluted in the protein extracts. Even though activation through the LR was connected with ROS, our initial analysis of the hypothalamus showed more differences in the CR KO line, which was why we restricted the additional tests to these groups.

As in the periphery, the role of neutrophils in the brain remained a primary interest in our present study. Recruitment of neutrophils to the brain has been recently highlighted as an important immune-to-brain communication pathway during systemic inflammation [55]. Since ARDS causes heavy recruitment of neutrophils into the lungs [97,98], we investigated if these cells could also play a role in lung-to-brain communication. Surprisingly, our initial results showed that the chemoattractant CXCL1 was elevated by n-3 PUFAs but such an increase was absent in both Rv receptor-deficient animals, although n-3 PUFAs did cause CR KO mice to produce significantly less CXCL1 than the Fat-Norm. These results were contradictory to what was expected since n-3 PUFAs reduce neutrophil infiltration and increase phagocytosis [9,128], but the precise dynamics remain to be further investigated in the future.

To better visualize neutrophil recruitment to the brain, further immunohistochemical analysis allowed us to not only quantify neutrophils but provided an opportunity to assess their recruitment to specific brain structures as well. Here, our results more accurately mirrored what has been shown previously [9,128,129]. Indeed, n-3 PUFAs eliminated the LPS-induced neutrophil peak observed in the WT-Norm mice, which was recovered by RvE1 receptor deficiency, suggesting a role for RvE1 in this response. As such, RvE1 has been previously shown to exhibit a significant impact on neutrophils during models of inflammation by attenuating LTB_4_s pro-inflammatory properties [21]. Moreover a combination treatment with aspirin and the RvE1 precursor 18R-HEPE into a dorsal air pouch in a mouse model of TNFα-induced local inflammation attenuated the recruitment of neutrophils to the inflammatory site [10,12]. Recently, we have even detected increased levels of 18-HEPE in supernatants of primary CVO cultures stimulated with LPS, highlighting 18-HEPE as a promising lead with the potential to treat neuroinflammation [130]. Although specific differences could not be detected between Norm groups, increased neutrophil recruitment to the BIF was observed in n-3 PUFA-enriched CR KO mice compared to Norm in our present study. Additionally, a stronger overall effect seen in mice lacking CR as opposed to the LR suggests that modulation of neutrophils during ARDS may occur primarily through this receptor. Indeed, the RvE1/CR axis is known to induce neutrophil apoptosis and phagocytosis [131,132].

Further attention was placed on the potential for lung-to-brain communication and the ability to modulate inflammation in the hypothalamus via cytokines. Interactions of cytokines with the brain during LPS-induced inflammation could contribute to the BBB’s loss of integrity and activation of brain endothelial cells [133]. In particular, IL-1β can cross the BBB as well as be produced locally by microglia [87,134,135]. We found that the mRNA expression of IL-1β and its antagonist IL-1ra were affected differently by n-3 PUFAs and RvE1 receptor KOs. The initial n-3 PUFA-dependent increase in basal levels of IL-1ra in CR KO mice may be related to microglial RvE1 expression. It is uncertain why this effect was limited to the CR KO mice in our study, but an early production of IL-1ra, such as from microglia [136] could limit LPS-induced inflammation through reduced emigration of neutrophils [137] and modulation of IL-1β pro-inflammatory activity [136,138,139]. To expand on our present mRNA expression data, further analysis is required to confirm the potential role of Rv signaling for brain IL-1β/IL-1ra action on protein levels.

Aside from cytokines and neutrophil trafficking, there are several influential signaling pathways that play a role during brain inflammation including NF-IL6, NFκB, and PPARγ [52,66,140,141]. All three of these pathways can be influenced by n-3 PUFA during models of inflammation [12,72,142,143]. In our present study, only minor impacts were found for the mRNA expression of NF-IL6 and PGC1α, a transcription coactivator that regulates signaling through PPARγ [143]. However, given that we have previously shown that NF-IL6 activation and unique lipid mediator production patterns both specifically occur in the OVLT of Fat mice during LPS-induced inflammation [72], we decided to pursue an additional analysis for NF-IL6. Even though RvE1 can limit activation of NFκB [12] and it is a prominent transcription factor for pro-inflammatory cytokines production [144], we did not determine any significant differences between groups in our present results and, thus, did not perform additional analysis for this pathway.

NF-IL6 has been shown to act as an early pro-inflammatory and late anti-inflammatory transcription factor during systemic inflammation and has been broadly used as a genomic brain cell activation marker [52,66]. More detailed results were obtained by spatiotemporal immunohistochemical assessment of NF-IL6 in the OVLT and the BIF. Interestingly, we did not observe significant differences in NF-IL6 immunoreactivity between groups nor by genetic n-3 PUFA enrichment at the level of the OVLT as we expected. However, subtle alterations due to n-3 PUFAs and RvE1 receptor deficiencies were present at the BIF suggesting some impact on brain cell activity. In particular, the Fat-CR KO mice may have experienced a more prolonged state of NF-IL6 activation, but high variability has to be taken into account. In addition, unchanged mRNA expression of the terminal enzyme for PGE2 production, namely mPGES, supports a limited impact of NF-IL6 known to be involved in the regulation of mPGES expression [66,145]. Overall, while some changes in brain cell activation marker NF-IL6 confirm modest brain alterations by LPS-induced ARDS, these minor changes again suggest that humoral signals did not play a major role in lung–brain signaling. 

Glia activation was also assessed given that systemic LPS stimulation has previously been demonstrated to robustly impact the function of astrocytes and microglia [133,134,146,147]. The release of cytokines, lipid mediators, or ROS by the glial cells plays an important role in regulating neuroinflammation [134,148,149,150]. Therapeutic techniques that modulate the activation of microglia or astrocytes could be useful in attenuating the resulting damage [151]. While we did not detect any significant changes in the microglial activation marker CD68 (Appendix A), GFAP, a marker for astrocytic activation, showed significant changes in CR KO mice. Interestingly, genetic n-3 PUFA enrichment increased GFAP expression in the CR KO mice compared to Norm. Additional immunohistochemical assessment of the hypothalamus at the level of the PVN confirmed that the astrocyte area was indeed increased in CR KO mice. Bousquet et al. (2011) have shown that Fat mice have reduced GFAP expression in a mouse model of Parkinson’s disease [152]. Since neuroinflammation can increase astrocyte expression of CR [122,153], it is possible that, through this receptor, RvE1 is able to modulate astrocyte activation, but these observations need to be further confirmed in future studies.

Important findings of the present manuscript also pertain to new insights into changes in lipid mediators in the lung and brain during LPS-induced ARDS. For lung inflammation, SPMs and especially EPA and its derivatives such as Rvs have been well studied during ARDS and the restoration of lung homeostasis [26,38,39,40]. It is known that the brain, on the other hand, contains high levels of PUFAs (25–30%) consisting mainly of DHA and AA [75]. In the present study, we were able to detect several lipid mediators and their derivatives. Most of these PUFAs were altered by genetic n-3 PUFA enrichment and the absence of RvE1 receptors in the lung and the brain. The knockdown of RvE1 receptors, especially LR, mainly led to an increase in PUFAs. Yet, RvE1 receptor knockdowns combined with genetic n-3 PUFA enrichment had an overall negative impact on PUFAs in the brain. In general, genetic n-3 PUFA enrichment in Norm and CR KO mice showed increased levels of PUFAs and their derivatives compared to the WT background counterparts, in particular at 0 h and 24 h p.i. This finding is supported by previous studies, which showed an increase in free n-3 and n-6 PUFAs in plasma as a general response to severe inflammation and infection [39,154,155]. Moreover, Mayer and colleagues (2003, 2009) revealed not only that DHA and EPA levels increase after infection or inflammation in human and murine plasma, but also showed further elevated levels of these PUFAs in Fat mice and patients treated with an n-3 PUFA emulsion [39,155]. Importantly, in a model of LPS-induced ARDS, Fat mice responded with lower production of inflammatory mediators and decreased neutrophil invasion to the alveoli [39]. Indeed, in our present study, we saw that the genetically enriched Fat mice showed higher amounts of several PUFAs normalized to WT mice at 0 h p.i. but further analyses remain to be performed to determine why more anti-inflammatory effects were not detectable in the lung and the hypothalamus/brain. 

As expected from previous studies [38,39,106], mice with genetic n-3 PUFA enrichment showed higher levels of n-3 PUFA EPA in the lung compared to Norm mice at 0 h p.i., but these levels decreased 24 h and 72 h p.i. Since EPA is converted to 18-HEPE and further to RvE1, higher amounts of these metabolites were found in mice with genetic n-3 PUFA enrichment compared to Norm mice at 24 h and 72 h p.i. Considering EPA is present at higher levels in the lung than the brain [73,74] and in accordance with the fact that the brain exhibited a higher percentage of DHA than EPA [75,76,77], a low abundance of 18-HEPE and undetectable amounts of RvE1 were expected in the brain. Nonetheless, RvE1 is a potent but unstable lipid mediator, which is rapidly converted to one of the inactive products and may, therefore, be difficult to detect [132,156,157]. These findings indicate high relevance of EPA and its derivatives in the lung while DHA and its derivatives are detected at higher levels in the brain. This may be of functional significance for lung–brain communication pathways.

The amount of DHA and especially its derivatives such as 17(S)-HDHA, 14(S)-HDHA, NPD1, and Mar1 were increased in the brain after 24 h p.i. in Norm and CR KO mice with n-3 PUFA enrichment. Of note, the 17S series DHA compounds 17(S)-HDHA and 14(S)-HDHA were proven to be potent regulators of leukocytes, reducing infiltration in vivo and they may also block cytokine production in glial cells [14]. These results are in line with other studies and show that NPD1 and Mar1 exhibit regulating effects in inflammation, which could also contribute to the resolution of inflammation in the brain [12,14,158,159,160,161,162].

In contrast to Norm and CR KO mice, genetic n-3 PUFA enrichment in LR KO mice did not exhibit as many elevated levels of PUFAs compared to the WT in the brain. AA and LTB_4_ were especially decreased in Fat-LR KO mice after 24 h p.i. Interestingly, different studies previously reported impaired recruitment of neutrophils accompanied by lowered inflammatory cytokine production in LR-deficient mice [32], suggesting a lower inflammatory response in these animals [90]. Such reduced inflammatory responses may also explain the absence of anti-inflammatory and pro-resolving lipid mediator responses in LR KO mice in the present study but further investigation to determine the precise role of PUFAs in LR KO mice is required. 

Limitations of the present study include technical challenges resulting in low numbers of samples per group. Overall, we have to acknowledge the screening character of the present study, which made use of the RRR principle of Russel and Burch [163] to perform our analyses on tissue that was harvested in collaboration from ongoing experiments for a different purpose. To reduce the amount of animals used for scientific purposes, we focused on LPS-stimulated mice. In this manner, parts of the lung were used for different experiments and we were not able to quantify neutrophils in the lung. In order to maximize the number of transgenic animals, males and females were both used for the experiment, which may have altered some of our assessments, as gender-specific alterations cannot be excluded. Previous studies have found that, compared to females, male mice have increased neutrophil influx and TNFα production during LPS-induced airway inflammation using a higher dose of LPS (50 µg) [164]. Due to low n numbers, we were not able to analyze by gender to rule out these effects in the present experiments. Future studies should expand on our present findings, for example, concerning metabolic pathways, assessing neutrophils in a more dynamic manner, as well as testing additional doses of LPS and the effects of gender. Moreover, different or more severe models of lung inflammation [165] such as a multi-hit model consisting of the intratracheal application of hydrochlorid acid and LPS [80] or simulating pulmonary infection with *Streptococcus pneumoniae* [166] may be suitable approaches to expand on the results of our present study with the aim to more precisely mimic the human situation such as the time course and components/subtypes of inflammation. Nonetheless, our present data identify an essential role for RvE1, i.e., its precursors and receptors, and neutrophils in lung-to-brain communication and explore the impact of lung inflammation, genetic n-3 PUFA enrichment, and Rv receptor deficiency on the brain.

## 4. Materials and Methods

### 4.1. Animals

The animal experiments were approved by local authorities (Regierungspräsideum Giessen: GI 20/10 Nr. G46/2017) and were performed in accordance with international legislation and the German Animal Welfare Act. All mice were obtained through in-house breeding in groups and kept under specific pathogen-free conditions on a 12 h day/night cycle at 22 ± 1 °C, relative humidity at ~50% with free access to standard laboratory chow and water. WT (C57BL/6NCrl) original breeding pairs were obtained from Charles River Laboratories (Sulzfeld, Germany), Fat (B6.129P2-Tg(CAG-fat-1)1Jxk) mice [51] were originally provided by Prof. J. X. Kang (Boston, MA, USA), CR KO (B6.129-CMKLR1 tm1_MBarnes) were originally provided by Prof. M. Barns (Takeda Cambridge, Cambridge, UK), and LR KO (B6.129S4_Ltb4r1<tm1Adl>/J) were originally purchased from Jackson Laboratory (Bar Harbor, ME, USA). Crossbreeding of Fat mice with CR KO mice obtained FAT-CR KO mice (B6-Tg(CAG-fat-1)1Jxk-CMKLR1 <tm1>Mbarnes/Ott) while crossbreeding of Fat mice with LR KO mice generated FAT-LR KO mice (B6-Tg(CAG-fat-1)1Jxk-Ltb4r1<tm1Adl>/Ott), which was performed in the lab of Prof. Dr. med. K. Mayer (Giessen, Germany). Animals of either sex aged 12 to 16 weeks (weight, ~23–25 g) were used for experiments.

### 4.2. Experimental Protocol 

Mice were anesthetized (0.25 mg/kg KGW Medetomidin/Domitor, 1 mg/mL, Elanco, Bad Homburg, Deutschland; diluted in saline, i.p.) and, in total, 50 µL (in 10–20–20 µL portions) LPS solution (10 µg Lipopolysaccharide, *Escherichia coli* O111:B4, in 50 µL normal saline/mouse) was instilled into the lung via a 22 G indwelling venous catheter through the trachea as previously described [167,168,169]. Intratracheal application of LPS mimics lung inflammation of the lower airways. Therefore, we have chosen such an application route to investigate lung–brain communication. Anesthesia was antagonized (0.25 mg/kg KGW Atipamezol/Antisedan, at 5 mg/mL, Zoetis, Berlin, Germany diluted in saline, s.c.) and after LPS application (0 h, 4 h, 24 h, 72 h, or 120 h), mice were sacrificed using the same anesthesia as stated above by blood withdrawal and organs were collected.

### 4.3. Immunohistochemistry

#### 4.3.1. General Protocol

To detect nuclear NF-IL6 immunoreactivity and neutrophil recruitment into the mouse brain, frozen brain sections were incubated with antibodies specific for NF-IL6 or MPO. On the day of the experiment, the brain sections were fixed in 100% ethanol (EtOH; Sigma-Aldrich, Munich, Germany) at −20 °C for 15 min then briefly dried with a hair dryer for approximately 5 s followed by three successive washes with PBS (only for sections stained with rabbit MPO and intercellular adhesion molecule 1 (ICAM1)). Other slides were air-dried at room temperature (rt) for 10 min. Next, sections were immersed in 2% paraformaldehyde (Sigma-Aldrich, Munich, Germany) diluted in PBS for 10 min at rt. After three successive washes with PBS, a blocking solution consisting of PBS, 10% normal donkey serum (NDS; Biozol Diagnostica, Eching, Germany), and 0.01% Triton X-100 (Sigma-Aldrich, Munich, Germany) was applied to the sections for 1 h at rt. Details of the primary antibody concentrations and combinations are shown in Appendix A. The sections were incubated with the primary antibodies overnight at 4 °C, followed by three successive washes with PBS, and visualization was achieved with Cy3- or Alexa-488-conjugated IgG also listed in Appendix A. Sections were counterstained with the nuclear 4.6-diamidin-2-phenylindol (DAPI) stain (MoBiTec, Göttingen, Germany) for 10 min using a 1:5000 dilution in PBS to clearly visualize the surrounding tissue and to demonstrate localization of NF-IL6 immunoreactivity. After a final three washes with PBS, a glass coverslip was used to cover the sections using glycerol/PBS solution (Citifluor, London, UK). The sections were stored at 4 °C until microscopic analysis was performed.

#### 4.3.2. NF-IL6

A rabbit antibody was used to detect nuclear NF-IL6 signals in the mouse brain in combination with von Willebrand factor (VWF) or MPO (see also Appendix A). Immunoreactivity in the BIF or OVLT was assessed per 20× field of view by semi-quantitative evaluation using a scale from 0 to 4, as previously performed [65].

#### 4.3.3. MPO

A rabbit or goat antibody was used to detect the neutrophil marker MPO in combination with ICAM1 or NF-IL6 (see also Appendix A). A component of myeloid cell granules MPO is commonly used in neutrophil staining [170]. Neutrophil recruitment into the BIF or OVLT was assessed per 10× field of view; all the neutrophils within the image were counted.

#### 4.3.4. GFAP

A guinea pig antibody was used to detect immunofluorescence of GFAP in the paraventricular nucleus (PVN) and evaluate the percentage of astrocytes area per image unit. To detect GFAP immunoreactivity in the mouse brain, frozen brain sections were incubated with a GFAP-specific antibody (see Appendix A). On the day of the experiment, the brain sections were immersed in 4% paraformaldehyde (Sigma-Aldrich, Munich, Germany) diluted in Tris-buffered saline (TBS) (Carl Roth GmbH + Co. KG, Karlsruhe, Germany) for 10 min at rt. After two successive washes with TBS, permeability was increased by applying 0.25% Triton X-100 (SERVA Electrophoresis GmbH, Heidelberg, Germany) diluted in PBS for 10 min at rt. Following three successive washes with TBS, a blocking solution consisting of TBS and 20% horse serum (Thermo Fisher Scientific Inc., Waltham, MA, USA) was applied to the sections for 1 h at rt. The primary GFAP antibody (see details in Appendix A) was prepared in a solution consisting of TBS, 5% horse serum, and 0.25% Triton X-100. The sections were incubated with the primary antibody overnight at 4 °C, followed by three successive washes with TBS, and visualization was achieved with Alexa 647-conjugated IgG (see details in Appendix A). Following three successive washes, the sections were counter-stained with Dapi (Carl Roth GmbH + Co. KG, Karlsruhe, Germany) for 5 min using a 1:500 dilution in TBS to visualize the surrounding tissue. After two final washes with TBS, a glass coverslip was used to cover the sections using ROTI^®^Mount, an aqueous mounting medium (Carl Roth GmbH+ Co. KG, Karlsruhe, Germany). The sections were stored at 4 °C until microscopic analysis was performed.

#### 4.3.5. Microscopic Analysis

Sections stained with NF-IL6 and MPO were analyzed using a conventional light/fluorescent Olympus BX50 microscope (Olympus Optical, Hamburg, Germany) equipped with the appropriate filter sets to detect the fluorescent conjugates used: Cy3, Alexa Fluor 488, and DAPI. Images were collected with a black-and-white Spot Insight camera (Diagnostic Instruments, Visitron Systems, Puchheim, Germany) and edited using Metamorph software (Version 7.7.5.0, Visitron Systems GmbH, Puchheim, Germany) to combine individual images into the RGB color figures. The brightness and contrast of collected images were adjusted uniformly within each experiment using Adobe Photoshop (Adobe System Inc., San Jose, CA, USA) for better visualization. Between 2 and 13 sections were imaged per animal with 1–5 animals per group used for analyses. In a few instances, due to damage to the brain or an insufficient number of slides, only one section could be collected for a structure. Additionally, as multiple brain slices were present per slide, occasionally mice have >5 sections per animal as a result of trying to obtain representative structures for other animals. Overall, some of these data were only used for descriptive purposes as statistical analyses were not possible due to low n-numbers. 

The immunofluorescence of GFAP was evaluated on snapshots taken with a Zeiss-Axio Observer 7 ACR immunofluorescence microscope (Carl Zeiss Microscopy Deutschland GmbH, Oberkochen, Germany). With the Apotome.2 module, several planes could be joined together in order to form a three-dimensional network, which had previously been recorded with the Colibri lights and installed incident light fluorescence. Between 1 and 3 sections were imaged per animal with 2–5 animals per group used for analyses.

### 4.4. Protein Carbonyl Content

The oxidation of protein results in the production of stable carbonyl groups, which are used as a measure of oxidative injury. Carbonyl content was determined with the Protein Carbonyl Content Assay Kit (Sigma MAK094, Merck KGaA, Darmstadt, Germany) according to the manufacturer’s protocol. In brief, hypothalamic tissue samples (20.0 ± 7.9 mg) were dissolved in dH2O, homogenized with a T10 basic Ultra-Turrax (IKA-Werke, Staufen, Germany), and sonicated for 10 min in a water bath at rt. Afterwards, the homogenates were centrifuged for 15 min at 13,000× *g* and 4 °C resulting in the protein-comprising supernatants (Heraeus Fresco 17; Thermo Fisher Scientific, Darmstadt, Germany). Next, the samples were treated with 1% streptozocin followed by successive incubation with 2,4-dinitrophenylhydrazine (DNPH), TCA, acetone, and guanidine. Absorbance was measured at 355 nm (FLUOstar OPTIMA reader, BMG Labtech, Ortenberg, Germany). To determine the amount of protein per sample, a Pierce BCA Protein Assay (Thermo Fisher Scientific, Darmstadt, Germany) was performed with 2 µL of the remaining solution. 

The carbonyl content was calculated as follows:$$CP\left(nmol\;carbonyl/mg\;protein\right)=\frac{\left(\frac{A_{355}}{6.364}\times 95\right)}{P}\times 1000$$
where 6.364 = extinction coefficient; 95 = total volume in well (µL); P = amount of protein in 95 µL sample; 1000 = conversion factor (µg to mg).

### 4.5. Protein Analysis

The pellets left over from the homogenization of the hypothalamic tissue were lysed in 0.25 M D-mannitol, 0.05 M Tris base, 1 mM EDTA, 1 mM EGTA, 1 mM DTT, and 1% Triton X-100 supplemented with protease and phosphatase inhibitor cocktail tablets (Roche Diagnostics, Mannheim, Germany). Total protein amounts were determined using the Pierce BCA Protein Assay Kit (Thermo Fisher Scientific, Darmstadt, Germany). Subsequently, 50 μg of protein in sample buffer (60 mM Tris HCl, 2% SDS, 10% glycerol, 5% β-mercaptoethanol, and 0.01% bromophenol blue) were loaded on 10% SDS-polyacrylamide gels. After gel electrophoresis and protein transfer to a polyvinylidine difluoride (PVDF) membrane (Roche Diagnostics, Mannheim, Germany), the membranes were incubated with the following antibodies: Vinculin (V9505, Mouse, 1:20,000, 116 kDa; Sigma, Munich, Germany), Catalase (sc-271803, Mouse, 1:10,000, 64 kDa; Santa Cruz, Heidelberg, Germany) and SOD-2 (NBP2-20535, Rabbit, 1:500, 25 kDa; Novus, Wiesbaden, Germany). Protein detection was realized using peroxidase-labeled secondary anti-mouse and anti-rabbit antibodies (1:1000, Vector Laboratories, Burlingame, CA, USA) and luminol-based HRP-Juice Plus (PJK GmbH, Kleinblittersdorf, Germany). The resulting chemiluminescence was imaged with a ChemiDoc XRS system (Bio-Rad Laboratories, Hercules, CA, USA). Densitometric protein quantification was performed using the Bio-Rad Image Lab Software 6.1 (Bio-Rad Laboratories, Hercules, CA, USA) according to the density of the bands (Vinculin 116 kDa, Catalase 64 kDa, SOD-2 25 kDa) in relation to WT 0 h group and normalized to Vinculin (loading control).

### 4.6. Cytokine Measurements

Lung and liver tissue of mice were homogenized in protease inhibitor cOmplete™ Mini (Roche, Penzberg, Germany) with Precellys 24 tissue homogenizer (Bertin Technologies SAS, Montigny-le-Bretonneux, France). After centrifugation (4 °C, 10 min, 14,000 rpm), the supernatant of these samples was used to perform the assay. The amount of protein in each sample was determined using DC Protein Assay (Bio-Rad Laboratories GmbH; Feldkirchen, Germany). Custom-made mouse magnetic bead-based multiplex assay obtained from R&D (R&D Systems, Minneapolis, MN, USA) were used to analyze levels of selected inflammatory mediators in lungs (CCL5/RANTES, CXCL1, CXCL5/LIX, GM-CSF, IL-1β, IL-6, IL-10, IL-17, TNFα; L144530) and liver (CCL5/RANTES, CXCL1, IL-1β, IL-6, IL-10, IL-17, CXCL5/LIX; L140916). To produce a comprehensive profile of cytokine levels in the tissue of the lung and liver, a luminex assay was again utilized. The cytokines measured in both tissue homogenates included CCL5/RANTES, CXCL1, CXCL5/LIX, IL-1β, IL-17, IL-10, and IL-6 while GM-CSF and TNFα were only measured in the lung. Together, these immune mediators are markers for inflammation, both initiate and modulate the sickness response and contribute to neutrophil recruitment and activation [80,101,102,103,104,171,172,173]. Through this analysis, it was possible to compare differences in inflammation in the infected lung tissue as well as changes in the liver response to systemic inflammation between groups [174]. The mouse magnetic bead-based multiplex assays were conducted according to the manufacturer’s protocol. The assay was performed with the Bio-Plex 200 instrument (Bio-Rad Laboratories GmbH; Feldkirchen, Germany) and analyzed with the Bio-Plex Manager Software 6.1 (Bio-Rad Laboratories GmbH; Feldkirchen, Germany). 

### 4.7. Eicosanoid Extraction and LC-MS/MS-Based Mass Spectrometric Analysis

The Eicosanoids and their derivatives were extracted from the lung and brain, and lipids from the lung were analyzed as previously described by high-pressure liquid chromatography-tandem mass spectrometry (LC-MS/MS) [56]. Accordingly, for the liquid chromatography, the 1100 Series capillary LC unit (Agilent Technologies Deutschland, Waldbronn, Germany) was used for samples of the lung and brain. Subsequently, the Esquire 3000+ ion trap mass spectrometer (Bruker Corporation, Billerica, MA, USA) was used for lung samples and the Daltonik amaZon SL (Bruker Corporation, Billerica, MA, USA) was used to examine the samples of the brain (settings: Appendix A). The characteristic mass spectrometric extracted ion chromatograms (EIC) traces were used for qualification and quantification (Appendix A).

### 4.8. Real Time (RT)-PCR

Total RNA from the collected hypothalamic tissue was isolated with Trizol (Thermo Fisher, Waltham, MA, USA) according to the manufacturer’s protocol. One microgram of total RNA was transcribed into DNA using 50 U murine leukemia virus (MULV) reverse transcriptase, 50 μM random hexamer, and 10 mM dNTP mix (Sigma-Aldrich, Munich, Germany) in a reaction volume of 20 μL. After the DNA was reverse transcribed quantitative real-time (RT) PCR was performed in duplicate using TaqMan Gene Expression Master Mix (Thermo Fisher, Waltham, MA, USA) and the respective primers. The cDNA quantities between different groups were normalized using the housekeeping gene glyceraldehyde 3- phosphate dehydrogenase (GAPDH; 4352339E-1009032, Applied Biosystems, Waltham, MA, USA) as a reference. Assay IDs for the analyzed genes are listed as follows: BDNF (Mm04230607_s1), Cat (Mm00437992_m1), CD163 (Mm00474091_m1), CD68 (Mm03047340_m1), CXCL1 (Mm04207460_m1), ELANE (Mm00469310_m1), GFAP (Mm01253033_m1), IL-1β (Mm00434228_m1), IL-1ra (Mm00446186_m1), IL-10 (Mm00439614_m1), IL-6 (Mm00446190_m1), mPGES (Mm00452105_m1), NF-IL6 (Mm00843434_s1), NfκBia (Mm00477798_m1), NOS2 (Mm00440502_m1), PGC-1α (Mm01208835_m1), SOCS3 (Mm00545913_s1), SOD1 (Mm01344232_g1), SOD2 (Mm01313000_m1), and TFAM (Mm00447485_m1). All primers were acquired from Thermo Fisher (Applied Biosystems) unless otherwise stated.

### 4.9. Data Analysis and Statistics

The fold change of relative quantity values normalized to WT-Norm 0 h controls was chosen for direct statistical analysis of gene mRNA expression. Parameters measured from tissue homogenate supernatants (cytokines, lipid mediators) were normalized to the protein amount of each sample before the fold change was normalized to WT-Norm 0 h and statistical analyses were used to account for differences in the protein amount caused by environmental factors between days of preparation. Cytokines and lipid mediators (normalized to tissue weight) were quantified by regression analysis of standard curve functions. In case of a not detectable cytokine or lipid mediator showing smaller quantities in the sample than the measurable range, these sample values were set to 0. In general, groups were analyzed by two-way ANOVA with the factors genetic n-3 PUFA enrichment (WT background, Fat background) and receptor deficiency (Norm, CR KO, LR KO). Main effects were reported, and Tukey post hoc testing was performed for single-group comparison. *p* < 0.05 was applied for statistical significance and mean ± SEM is displayed in the figures.

#### 4.9.1. Evaluation of Cytokine Measurements

The concentrations of cytokines in the lung and liver tissue were determined according to the manufacturer’s instructions for the assay using the standards provided. WT and Fat groups were compared to each other within each group: Norm, CR KO, and LR KO. Additionally, the WT and Fat mice from CR KO and LR KO groups were compared to the respective Norm groups but not to each other. Each cytokine was analyzed by two-way ANOVA with the factors condition (Rv receptor deficiency) and treatment (n-3 PUFA enrichment) and a Tukey post hoc test. Outlier analyses were performed using the ROUT method [175] and values were excluded (Liver-IL-10 at 24 h: 1 WT-CR KO/CXCL1 at 24 h: 1 WT-Norm, 1 Fat-CR KO; 72 h: 1 WT-Norm/IL-17 at 24 h: 1 WT-CR KO). All statistical analyses were performed using the GraphPad Prism (Version 7; GraphPad Software, San Diego, CA, USA).

#### 4.9.2. Evaluation of RT-PCR

The relative quantification (RQ) was determined based on the cycle threshold (ct) defined when amplification exceeds that of the background signal. Once the ct has been reached, the cDNA amplification becomes exponential and will double with each cycle and can be applied for relative quantification. The RQ values were used to compare target genes from the q-RT-PCR analysis. RQ values were normalized to the lowest expression in the control group (0 h WT-Norm) for each target gene and assigned a value of 1. All other values were shown as a multiple of the expression [176]. The relative expression from the control sample (given a value of 1) from the 0 h time point for each group was transformed and reported as fold change normalized to the WT-Norm control mice from the 0 h time point. WT and Fat groups were compared to each other within each group: Norm, CR KO, and LR KO. Additionally, the WT and Fat mice from CR KO and LR KO groups were compared to the respective Norm group but not to each other. RT-PCR for target genes was analyzed for all groups in the experiment on one day to limit variation. Analyses were performed using a two-way ANOVA with the factors condition (Rv receptor deficiency) and treatment (n-3 PUFA enrichment) and a Tukey post hoc test for both. Outlier analyses were performed using the ROUT method [175] and values were excluded (SOD1 at 72 h: 1 WT-Norm/Elane at 0 h: 1 Fat-Norm; 24 h: 1 Fat-CR KO, 1 Fat-LR KO; 72 h 1 Fat-CR KO/IL-10 at 24 h: 1 Fat-CR KO/IL-1β at 0 h: 1 WT-Norm/IL-1ra at 0 h: 1 WT-Norm, 1 WT-CR KO/NF-IL6 at 0 h: 1 Fat-Norm/mPGES at 72 h: 1 Fat-Norm). All statistical analyses were performed using the GraphPad Prism (Version 7; GraphPad Software, San Diego, CA, USA). For samples where a value could not be determined they were removed from the data set (Elane at 0 h: 1 WT-Norm, 2 Fat-LR KO; 24 h: 1 Fat-Norm; 72 h: 1 WT-Norm/IL-10 at 0 h: 3 WT-Norm, 3 Fat-Norm, 1 WT-CR KO, 2 Fat-CR KO, 1 Fat-LR KO; 24 h: 4 WT-Norm, 1 Fat-Norm, 1 WT-CR KO, 2 WT-LR KO, 1 Fat-LR KO; 72 h: 1 WT-Norm, 1 Fat-Norm, 2 WT-CR KO, 1 Fat-CR KO, 2 WT-LR KO, 1 Fat-LR KO).

#### 4.9.3. Evaluation of Western Blot Protein Analysis

Protein levels were obtained by densitometric quantification of the Western blot data and normalization to Vinculin (loading control). The data are presented as a fold of the control group (Norm WT 0h). WT and Fat groups were compared to each other within each group: Norm and CR KO (n = 4–5). Multiple comparisons were performed by ANOVA and Tukey post hoc test. The statistical analyses were carried out using GraphPad Prism (Version 7; GraphPad Software, La Jolla, CA, USA).

#### 4.9.4. Evaluation of Immunohistochemistry

In all instances where immunofluorescence was used, brain sections were exposed to the optimal wavelength of light and an image was collected. WT and Fat groups were compared to each other within each group: Norm, CR KO, and LR KO. Additionally, the WT and Fat mice from CR KO and LR KO groups were compared to the respective Norm group but not to each other. Analyses were performed using a two-way ANOVA with the factors condition (Rv receptor deficiency) and treatment (n-3 PUFA enrichment) and a Tukey post-hoc test for both. Outlier analyses were performed using the ROUT method [175]. All statistical analyses were performed using the GraphPad Prism (Version 7; GraphPad Software, San Diego, CA, USA).

##### NF-IL6 and MPO

For immunofluorescence analyses of NF-IL6 and MPO, either quantitative or semi-quantitative measurements of fluorescent signals were made. When quantitative evaluation was used, i.e., for MPO analysis, the nuclear signals were counted. Groups stained for MPO had n = 3 mice unless stated otherwise here, ‘n’ are reported as follows as OVLT, BIF: 0 h Fat-Norm = 3, 5/4 h WT-Norm = 2, 2; Fat-Norm = 5, 5; WT-LR KO = 4, 4/24 h WT-Norm = 3, 4; Fat-Norm = 5, 5; WT-CR KO = 4, 4; Fat-CR KO = 4, 4; WT-LR KO = 5, 5; Fat-LR KO = 4, 4/72 h WT-Norm = 3, 4; Fat-Norm = 5, 5; WT-CR KO = 4, 4; Fat-CR KO = 3, 4; WT-LR KO = 5, 5; Fat-LR KO = 3, 4/120 h Fat-Norm = 4, 5; Fat-CR KO = 2, 2; WT-LR KO = 3, 4; Fat-LR KO = 4, 4.

During semi-quantitative evaluation, i.e., NF-IL6 analysis, the level of nuclear signals present was scored for each section on a scale from 0 to 4: 0 = none, 1 = single or very few, 2 = low density, 3 = moderate density, 4 = high density; ¼ scores were assigned when slightly more or less than average signals were present but not enough to move up or down an entire level, i.e., 1.25 or 1.75 where a score of 1.25 has slightly more signal than 1 and 1.75 has slightly less signal than 2. Groups stained for NF-IL6 had n = 3 mice unless stated otherwise here, ‘n’ are reported as follows as OVLT, BIF: 0 h Fat-Norm = 4, 5/4 h WT-Norm = 1, 2; Fat-Norm = 5, 4/24 h WT-Norm = 4, 4; Fat-Norm = 5, 4; WT-CR KO = 4, 4; Fat-CR KO = 4, 4; Fat-LR KO = 4, 4/72 h WT-Norm = 4, 4; Fat-Norm = 5, 4; WT-CR KO = 4, 4; Fat-CR KO = 3, 4/120 h Fat-Norm = 3, 4; WT-CR KO = 2, 3; Fat-CR KO = 2, 2; WT-LR KO = 3, 4.

##### GFAP

For immunofluorescence analysis of GFAP, the evaluation of the images was carried out using the program ZEN 3.6 (Carl Zeiss Microscopy GmbH, Oberkochen, Germany). The measurements were carried out with the module “Intellesis Segmentation” and the percentage of astrocytes area per image unit was determined.

#### 4.9.5. Evaluation of LC-MS/MS

The mass-spectrometric raw data were quantified as previously described [56]. All statistical analyses were performed using the GraphPad Prism (Version 7; GraphPad Software, San Diego, CA, USA).

## 5. Conclusions

In our hands, even with some of our present results being descriptive, we revealed that LPS-induced ARDS increased inflammatory signaling in the lung and the hypothalamus where low humoral signaling and immune cell trafficking to the brain potentially contributed to immune-to-brain communication (Figure 10). A deficiency in RvE1 receptors as well as enhanced production of n-3 PUFAs was able to sufficiently alter the profile of several inflammatory mediators, modify GFAP immunoreactivity (for CR deficiency), and alter lipid mediators in the brain. In conclusion, we have presented evidence that genetic enrichment of n-3 PUFAs and RvE1 receptor deficiencies can alter the lipid profile in the brain during lung inflammation. In particular, acting through CR, RvE1 may be able to modulate communication to some degree through the humoral pathway and neutrophil recruitment. Moreover, we suggest a role for n-3 PUFAs to modulate neutrophil recruitment to the brain during lung inflammation. Overall, n-3 PUFAs and, in particular, SPMs, i.e., NPD1/PDX or Mar1 and SPM precursors such as 18-HEPE, 17(S)-HDHA, or 14(S)-HDHA, respectively, may represent good targets for future investigations to modulate inflammation in individuals suffering from or at risk for developing brain inflammation as a result of ARDS.

## Figures and Tables

**Figure 1 ijms-24-13524-f001:**
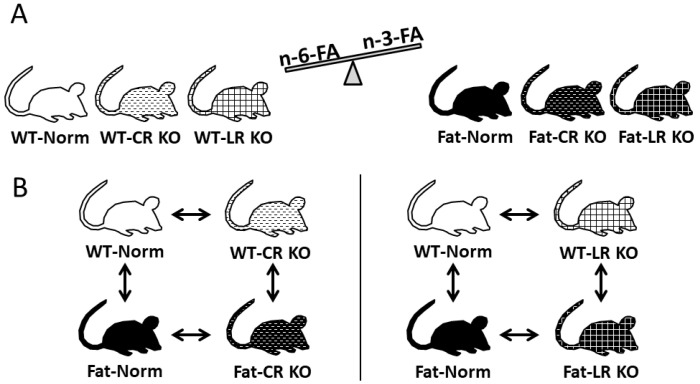
Schematic for genetically modified mouse strains. The breeding strategy outlines the mice used in the experiments (**A**). Genetically enriched n-3 fatty acid producing *fat-1* mice (Fat; black) were compared to wild-type control mice (WT; white). Both WT and Fat mice were cross bred with knockout (KO) mice deficient in the resolvin receptor chemerin receptor 23 (CR KO; spotted) or leukotriene B4 receptor (LR KO; checkered) to generate KO mice on WT and Fat backgrounds. Mice that maintained intact resolvin receptors were designated as Norm. Comparisons were made between Norm and receptor-deficient mice on both WT and Fat backgrounds but CR KO and LR KO mice were not compared to each other (**B**).

**Figure 2 ijms-24-13524-f002:**
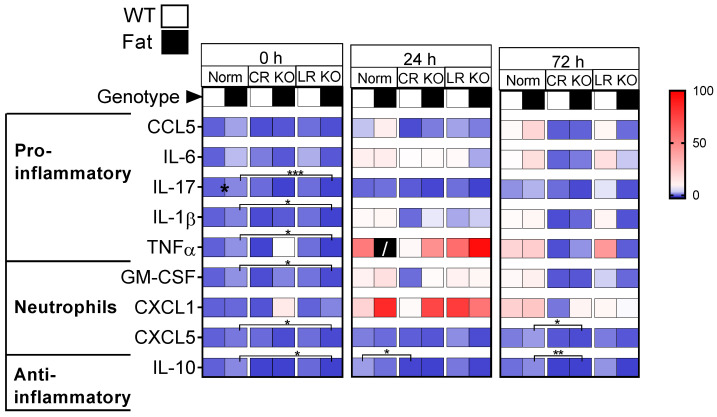
Changes in mediators of lung inflammation following intratracheal LPS-induced ARDS. Multiplex cytokine measurements from homogenized lung tissue for pro-inflammatory cytokines: chemokine (C-C motif) ligand 5 (CCL5), interleukin (IL)-6, IL-17, IL-1β, and tumor necrosis factor (TNF)α; neutrophil chemoattractants: granulocyte-macrophage colony-stimulating factor (GM-CSF), chemokine (C-X-C motif) ligand (CXCL)1 and CXCL5; and the anti-inflammatory cytokine: IL-10. Mice deficient in chemerin receptor 23 (CR KO) or leukotriene B4 receptor (LR KO) as well as unmodified mice (Norm) bred on wild-type (WT) or transgenic omega-3 (n-3) synthesizing *fat-1* (Fat) background received an intratracheal (i.t.) instillation with lipopolysaccharide (LPS, 10 µg) and were sacrificed at 0 h, 24 h, or 72 h p.i. Impacts of n-3 PUFAs and the RvE1 receptors CR and LR are shown at each time point to different extents. Indeed, significant alterations from n-3 PUFAs and the CR are detected for IL-10. The average fold change for each sample set was normalized to WT-Norm 0 h and is presented as a value on a heat map. A black box with a white diagonal line indicates a value that exceeds the scaled range. n = 3 per group for all groups except Fat-CR KO 0 h; n = 2; therefore, this group was excluded from statistical analyses but visually kept as descriptive information. * *p* < 0.05, ** *p* < 0.01, *** *p* < 0.001.

**Figure 3 ijms-24-13524-f003:**
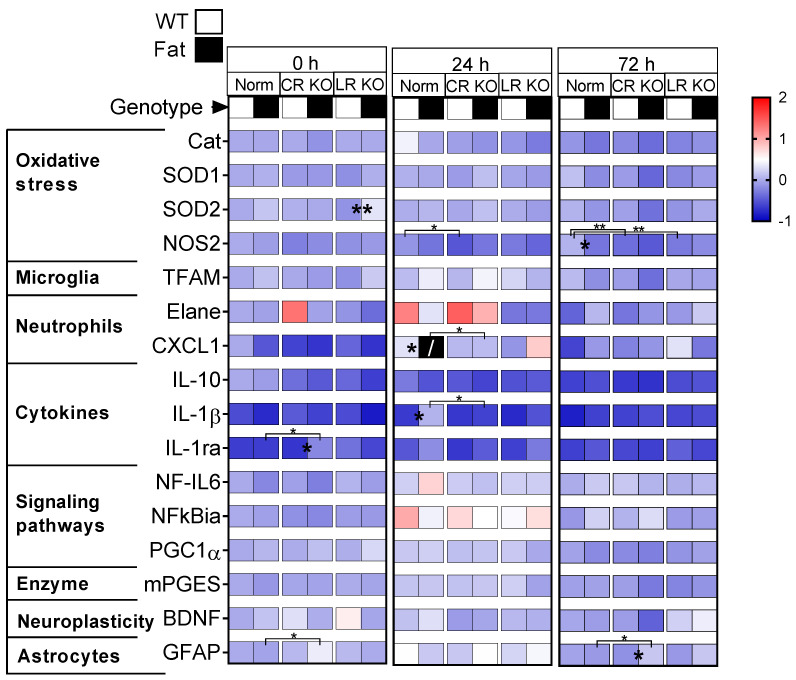
Changes in hypothalamic inflammatory mediator mRNA expression following intratracheal LPS-induced ARDS. The hypothalamus was analyzed for expression of markers for oxidative stress: catalase (Cat), superoxide dismutase (SOD)1, SOD2 and nitric oxide synthase (NOS2); mitochondrial biogenesis: transcription factor A, mitochondrial (TFAM); neutrophil trafficking: elastase, neutrophil expressed (Elane), and chemokine (C-X-C motif) ligand (CXCL)1; cytokines: interleukin (IL)10, IL-1β, and IL-1ra; signaling pathways: nuclear factor (NF)-IL6, NFκB via IκBα (ia), and peroxisome proliferator-activated receptor-α coactivator (PGC)1α; enzymes involved in prostaglandin expression: microsomal prostaglandin E synthase (mPGES); neuroplasticity: brain-derived neurotrophic factor (BDNF); and astrocyte activation: glial fibrillary acidic protein (GFAP). Mice deficient in chemerin receptor 23 (CR KO) or leukotriene B4 receptor (LR KO) as well as unmodified mice (Norm) bred on wild-type (WT) or transgenic omega-3 (n-3) synthesizing *fat-1* (Fat) background received an intratracheal (i.t.) instillation with lipopolysaccharide (LPS, 10 µg) and were sacrificed at 0 h, 24 h, or 72 h p.i. An impact of n-3 PUFAs and the RvE1 receptors CR and LR are evident at each time point but for different markers. The average fold change for each sample set was normalized to WT-Norm 0 h and presented as a value on a heat map. A black box with a white diagonal line indicates a value that exceeds the scaled range. n = 3–6 per group after outlier exclusions and undetermined values. Only IL-10 had fewer samples (n = 1–5) for which statistics were not performed but visually kept as descriptive information. * *p* < 0.05, ** *p* < 0.01.

**Figure 4 ijms-24-13524-f004:**
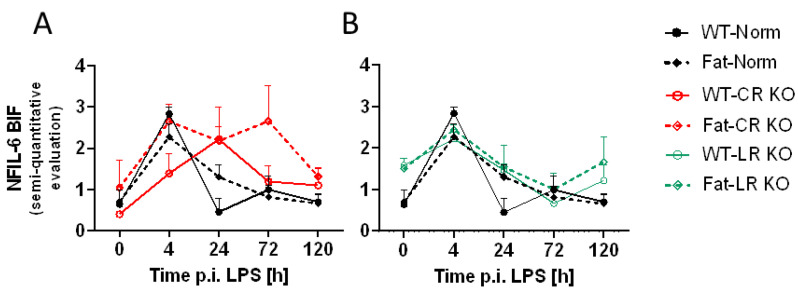
NF-IL6 immunoreactivity in Norm, CR KO, and LR KO mice at the level of the BIF over time after intratracheal LPS-induced ARDS. Sections of the brain analyzed at the level of the bifurcation (BIF) for nuclear factor interleukin 6 (NF-IL6) immunoreactivity on a scale ranging from 0 to 4 (**A**,**B**). Unmodified mice (Norm) were compared to mice deficient in chemerin receptor 23 (CR KO; (**A**)) or leukotriene B4 receptor (LR KO; (**B**)) bred on a wild-type (WT) or transgenic omega-3 (n-3) synthesizing *fat-1* (Fat) background received an intratracheal (i.t.) instillation with lipopolysaccharide (LPS, 10 µg) and were sacrificed at 0 h, 4 h, 24 h, 72 h, and 120 h p.i. Sections were analyzed by immunofluorescent staining using an antibody for NF-IL6. Immunoreactivity was assessed using semi-quantitative evaluation. n = 2–5 per group. Statistics were not performed due to low ‘n’ numbers.

**Figure 5 ijms-24-13524-f005:**
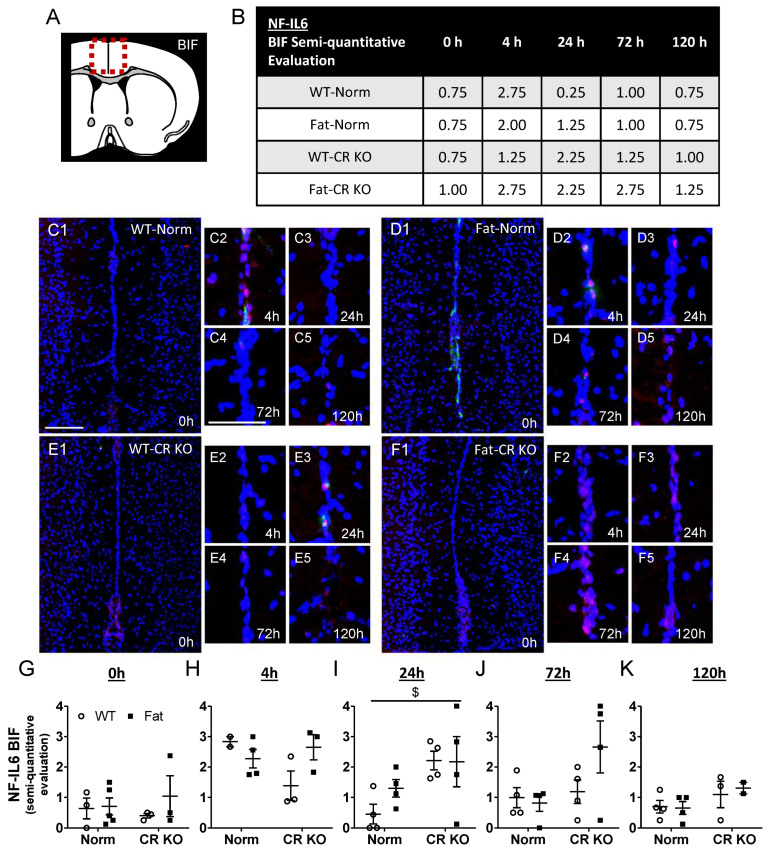
Intratracheal LPS-induced ARDS NF-IL6 immunoreactivity at the level of the BIF in Norm compared to CR KO mice is significantly altered at 24 h p.i. Sections of the brain were analyzed at the level of the bifurcation (BIF; (**A**) indicated by dashed square) by semi-quantitative evaluation for nuclear factor interleukin 6 (NF-IL6; red, (**C**–**F**)) immunoreactivity on a scale ranging from 0 to 4 (**B**). Unmodified mice (Norm; (**C**)) were compared to mice deficient in the chemerin receptor 23 (CR KO; (**E**,**F**)) bred on a wild-type (WT; (**C**,**E**)) or transgenic omega-3 (n-3) synthesizing *fat-1* (Fat; (**D**,**F**)) background received an intratracheal (i.t.) instillation with lipopolysaccharide (LPS, 10 µg) and were sacrificed at 0 h, 4 h, 24 h, 72 h, and 120 h p.i. WT-/Fat-Norm groups were compared to WT-/Fat-CR KO groups at each time point (**G**–**K**). At 24 h p.i with LPS CR deficiency increased NF-IL6 immunoreactivity compared to Norm. Von Willebrand factor (green; (**C**–**F**)) depicts brain vasculature. DAPI (blue; (**C**–**F**)) visualizes the surrounding tissue. n = 2–5 per group. Statistics were only performed when n = >3 per group. $ main effect Norm vs. CR KO. Scale bar in (**C1**) = 100 μm and is representative of (**C1**,**D1**,**E1**,**F1**); Scale bar in (**C4**) = 50 µm and is representative of (**C2**–**C5**,**D2**–**D5**,**E2**–**E5**,**F2**–**F5**). $ *p* < 0.05.

**Figure 6 ijms-24-13524-f006:**
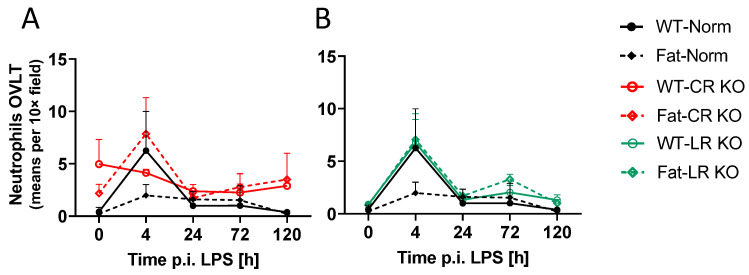
Neutrophil recruitment in Norm, CR KO, and LR KO mice at the level of the OVLT over time after intratracheal LPS-induced ARDS. Sections of the brain were analyzed at the level of the vascular organ of lamina terminalis (OVLT) for neutrophil trafficking to the brain (**A**,**B**). Unmodified mice (Norm) were compared to mice deficient in chemerin receptor 23 (CR KO; (**A**)) or leukotriene B4 receptor (LR KO; (**B**)) bred on wild-type (WT) or transgenic omega-3 (n-3) synthesizing *fat-1* (Fat) background received an intratracheal (i.t.) instillation with lipopolysaccharide (LPS, 10 µg) and were sacrificed at 0 h, 4 h, 24 h, 72 h, and 120 h p.i. Sections were analyzed by immunofluorescence staining using an antibody for myeloperoxidase. Images were taken and neutrophils were counted per 10× field of view. n = 2–5 per group. Statistics were not performed due to low ‘n’ numbers.

**Figure 7 ijms-24-13524-f007:**
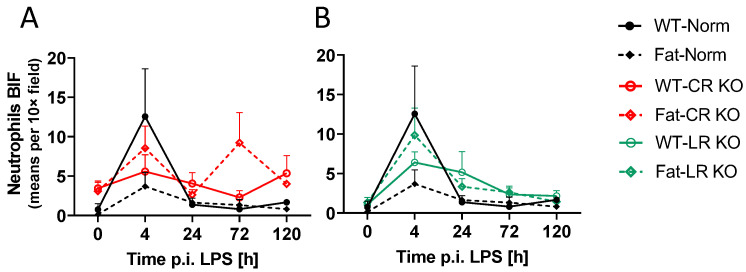
Neutrophil recruitment in Norm, CR KO, and LR KO mice at the level of the BIF over time after intratracheal LPS-induced ARDS. Sections of the brain were analyzed at the level of the bifurcation (BIF) for neutrophil trafficking to the brain (**A**,**B**). Unmodified mice (Norm) were compared to mice deficient in chemerin receptor 23 (CR KO; (**A**)) or leukotriene B4 receptor (LR KO; (**B**)) bred on a wild-type (WT) or transgenic omega-3 (n-3) synthesizing *fat-1* (Fat) background received an intratracheal (i.t.) instillation with lipopolysaccharide (LPS, 10 µg) and were sacrificed at 0 h, 4 h, 24 h, 72 h, and 120 h p.i. Sections were analyzed by immunofluorescence staining using an antibody for myeloperoxidase. Images were taken and neutrophils were counted per 10× field of view. n = 2–6 per group. Statistics were not performed due to low ‘n’ numbers.

**Figure 8 ijms-24-13524-f008:**
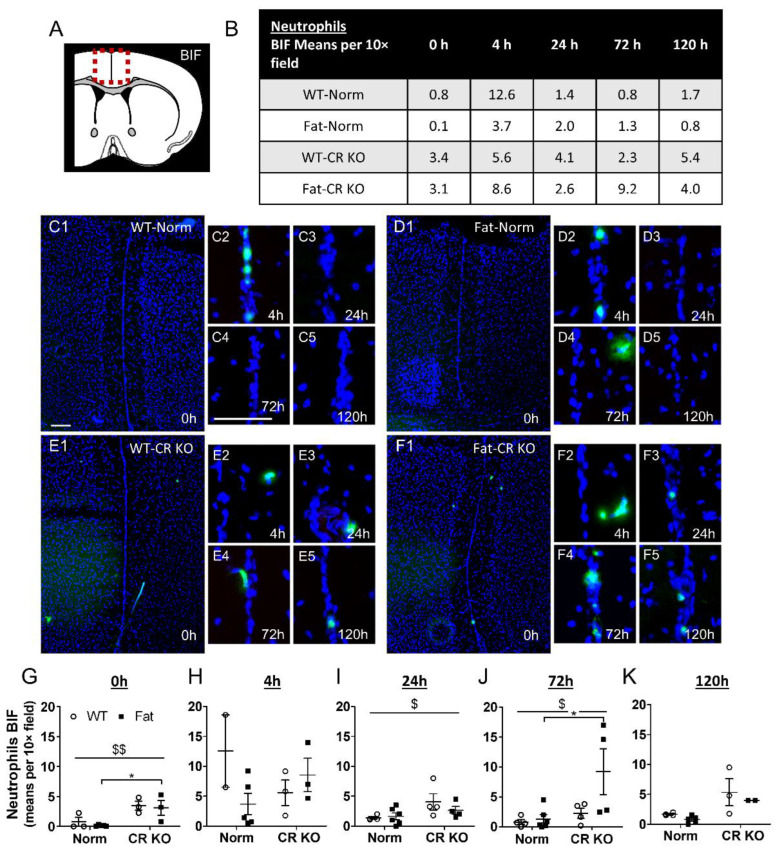
LPS-induced ARDS neutrophil recruitment at the level of the BIF in Norm compared to CR KO mice is significantly altered at 0 h, 24 h, 72 h, and 120 h p.i. Sections of the brain were analyzed at the level of the bifurcation (BIF; (**A**) indicated by the dashed square) and neutrophils (green, (**C**–**F**)) were counted per 10× field of view (**B**). Unmodified mice (Norm; (**C**)) were compared to mice deficient in the chemerin receptor 23 (CR KO; (**E**)) bred on wild-type (WT; (**C**,**E**)) or transgenic omega-3 (n-3) synthesizing *fat-1* (Fat; (**D**,**F**)) background received an intratracheal (i.t.) instillation with lipopolysaccharide (LPS, 10 µg) and were sacrificed at 0 h, 4 h, 24 h, 72 h, and 120 h p.i. WT-/Fat-Norm groups were compared to WT-/Fat-CR KO groups at each time point (**G**–**K**). At 0 h, 24 h, 72 h and 120 h p.i with LPS CR KO had altered neutrophil recruitment compared to Norm. DAPI (blue; (**C**–**F**)) visualizes the surrounding tissue. n = 2–6 per group. Statistics were only performed when n = >3 per group. $ main effect Norm vs. CR KO. Scale bar in (**C1**) = 100 µm and is representative of (**C1**,**D1**,**E1**,**F1**); Scale bar in (**C4**) = 50 μm and is representative of (**C2**–**C5**,**D2**–**D5**,**E2**–**E5**,**F2**–**F5**). * *p* < 0.05, $ *p* < 0.05, $$ *p* < 0.01.

**Figure 9 ijms-24-13524-f009:**
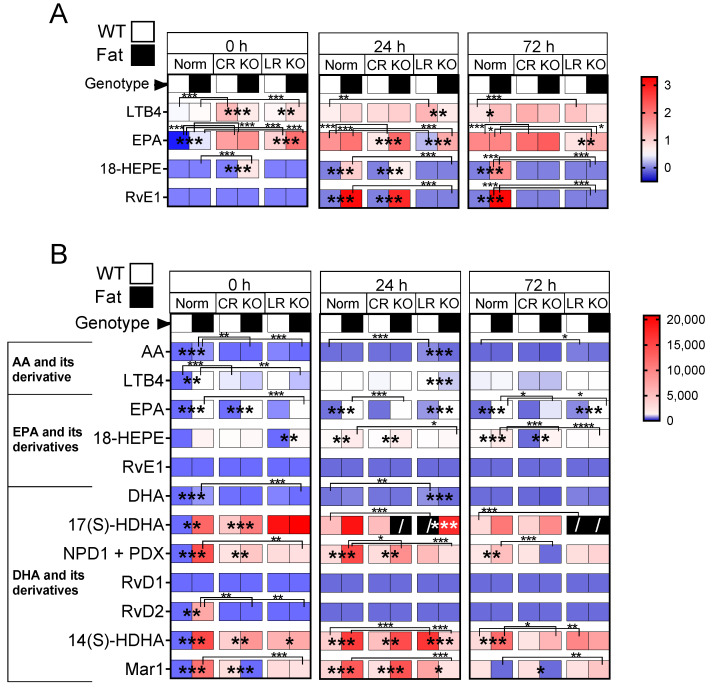
Differences in the lipid mediators in the (**A**) lung and (**B**) brain following intratracheal LPS-induced ARDS. Lipid mediators were measured from homogenized lung (**A**) and brain tissue (**B**) with LC-MS/MS. Mice deficient in chemerin receptor 23 (CR KO) or leukotriene B4 receptor (LR KO) as well as unmodified mice (Norm) bred on wild-type (WT) or transgenic omega-3 (n-3) synthesizing *fat-1* (Fat) background received an intratracheal (i.t.) instillation with lipopolysaccharide (LPS, 10 µg) and were sacrificed at 0 h, 24 h, or 72 h p.i. Notably, impacts of genetic n-3 PUFA enrichment and RvE1 receptor deficiencies (CR or LR) are observed in the lung in leukotriene B4 (LTB_4_), eicosapentaenoic acid (EPA), 18R-hydroxy eicosapentaenoic acid (18-HEPE) and resolvin E1 (RvE1) levels. In the hypothalamus, several polyunsaturated fatty acids (PUFA) and their derivatives including arachidonic acid (AA), LTB_4_, EPA, 18-HEPE, RvE1, docosahexaenoic acid (DHA), 17(S)-hydroxy docosahexaenoic acid (17(S)-HDHA), neuroprotectin D1 (NPD1) + protectin DX (PDX), resolvin (RV)D1, RvD2, 14(S)-hydroxy docosahexaenoic acid (14(S)-HDHA), and maresin 1 (Mar1) were altered. The average fold change for each sample set is normalized to WT-Norm 0 h and is presented as a value on a heat map. A black box with a white diagonal line indicates a value that exceeds the scaled range. n = 3–6 per group after outlier exclusions and undetermined values. n = 5 per group for all groups except at the 0 h time point Fat-CR KO and at 72 h WT-Norm (n = 4). * *p* < 0.05, ** *p* < 0.01, *** *p* < 0.001.

**Figure 10 ijms-24-13524-f010:**
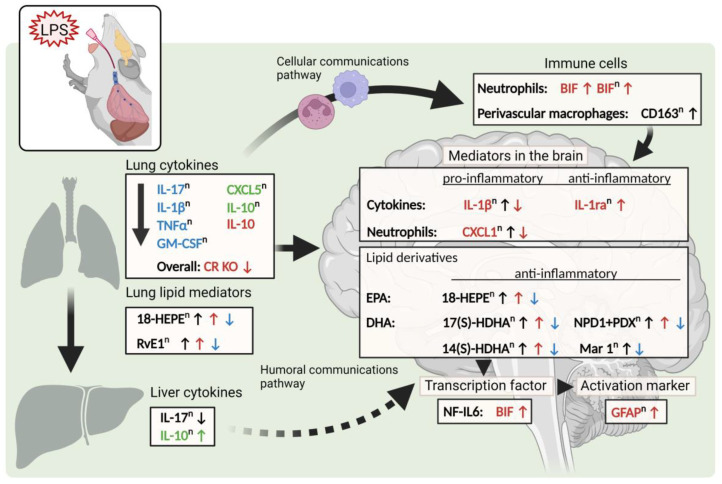
Influence of n-3 PUFAs and RvE1 receptor knockouts on lung and brain inflammation during ARDS. Schematic summary of the main findings: The inflammatory response in the lung, liver, and hypothalamus following intratracheal instillation with lipopolysaccharide (LPS, 10 μg) induced ARDS includes the expression of inflammatory mediators, immune cells, lipid derivatives, transcription factors, and activation markers at 0 h, 24 h, and 72 h p.i. with LPS. Further analyses of neutrophil trafficking and nuclear factor (NF)-IL6 immunoreactivity in two brain structures, namely, the bifurcation (BIF) and vascular organ of lamina terminalis (OVLT) were carried out at the same time points as well as at 4 h and 120 h p.i. Changes are shown compared to unmodified controls (Norm) bred on a wild-type (WT) background and were color-coded for effects in resolvin (Rv) receptor-deficient groups: chemerin receptor 23 (CR) KO (red), leukotriene B4 receptor (LR) KO (blue) or both KOs (CR KO and LR KO; green). Comparisons between Norm and KO mice on a transgenic n-3 synthesizing *fat-1* (Fat) background are identified with an ‘n’ and include comparing Fat-Norm to WT-Norm (black). Arrows in the corresponding colors signify an increase (up arrow ↑) or decrease (down arrow ↓) for that genotype. A large arrow within the lung cytokine box indicates that all cytokines were decreased in their respective comparisons. Mediators in the brain were primarily altered in CR KO groups but black arrows also indicate effects when comparing Fat-Norm to WT-Norm. Lipid mediators are written in black due to extensive effects between groups and arrows indicate genotype-specific alterations. Larger arrows outside of text boxes show a pathway for immune-to-brain communication, when they are dashed it indicates a weak source of communication with the brain. Arrow heads (▼) were used to outline possible pathways of activation in the brain. Abbreviations: interleukin (IL), granulocyte-macrophage colony-stimulating factor (GM-CSF), chemokine (C-X-C motif) ligand (CXCL), cluster of differentiation (CD), hydroxy eicosapentaenoic acid (HEPE), hydroxy docosahexaenoic acid (HDHA), neuroprotection D1 (NPD1), protectin DX (PDX), maresin 1 (Mar1) and glial fibrillary acidic protein (GFAP).

## Data Availability

The data presented in this study are contained within the article and Appendix A or are available upon reasonable request from the corresponding author.

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
