# Peer review of "n-3 Polyunsaturated Fatty Acids Modulate LPS-Induced ARDS and the Lung–Brain Axis of Communication in Wild-Type versus Fat-1 Mice Genetically Modified for Leukotriene B4 Receptor 1 or Chemerin Receptor 23 Knockout"

_ijms, 2023, doi:10.3390/ijms241713524_

Round 1
Reviewer 1 Report
The manuscript was written with a clear demonstration of the method and design of the study and a well-written results section.
There was almost no need to improve the English language.
Author Response
The manuscript was written with a clear demonstration of the method and design of the study and a well-written results section.
Response: We thank the referee for the positive evaluation of our manuscript. The comments of all three referees helped to improved our revised manuscript.
Reviewer 2 Report
The role of n-3 PUFA in brain during LPS induced ARDS mice was studied in this manuscript. The n-3 enriched Fat and deficient Rv receptors (CR KO or LR KO mice) were used in this study.
Comments:
1. Figure 3, from the heat map color, the average fold changes of TNFa and CXCL1 look like significantly. However, the authors didn’t get statistical difference. Is it because of the low number of mice in each group (n=2 or 3) or variability matter? The mice used in this study are either male or female, however, males exhibited a greater degree of neutrophilic influx and TNFa than females in LPS induced ARDS.( J Immunol. 2006 Jul 1; 177(1): 621–630.) The gender should be considered.
2. The values and the florescence photos are not match. Figure 7B, the BIF values of 4h and 72h in Fat-CR KO mice are both 2.75, however, from the immunoreactivity photos, red florescence is much higher at 72h compare to 4h. Figure 10B, from the immunoreactivity photos, WT-Norm 0h and FAT-CRKO 0h showed no difference, the value are 0.8 VS 3.1? And the peak value is 12.6 at WT-Norm 4h, however, florescence density isn’t higher than FAT-CRKO at 4h and 72h.
3. Discussion part is too long. Macrophage polarization is not related to this study.
Author Response
Comments:
- Figure 3, from the heat map color, the average fold changes of TNFa and CXCL1 look like significantly. However, the authors didn’t get statistical difference. Is it because of the low number of mice in each group (n=2 or 3) or variability matter? The mice used in this study are either male or female, however, males exhibited a greater degree of neutrophilic influx and TNFa than females in LPS induced ARDS.( J Immunol. 2006 Jul 1; 177(1): 621–630.) The gender should be considered.
Response: We apologize for not being clear enough about the reasons why the differences in the Fig 3 were not significant. While we agree that the fold change on the heatmap may visually appear greater in comparison, as already suggested by this referee, due to a small sample size (n=2-3) and variability between samples we could not always detect differences between groups, such as TNFa and CXCL1. In future studies, increasing n-numbers will be important to further assess functional aspects in this model. To address this comment, we have now added an additional statement to the results section.
Page 5, lines 209-210 “Given the low numbers of animals per group and the degree of variation, only large effect sizes were determined and some low effect size differences may not always have been detected.“
Moreover, we agree that gender is an important component that needs to be considered. Since the experiment relied on the three R principle (Replacement, Reduction, and Refinement), we collected the tissue from a pre-existing experiment. As in previous studies with this complex mouse model, male and female mice were pooled for analyses [1]. We are aware of gender differences that may or may not alter the response to stimuli like LPS. However, previous experiments did use both gender in the same model revealing new insights into the role of n-3 PUFA during LPS-induced lung inflammation [1]. Moreover, the small group numbers in the present study do not enable meaningful analyses by gender. Nonetheless, to address this important limitation, we now have added the suggested study on neutrophils and gender in the “limitations section” of the discussion.
Page 28, lines 1126-1130 “Previous studies have found that, compared to females, male mice have increased neutrophil influx and TNFa production during LPS-induced airway inflammation using a higher dose of LPS (50 µg). Due to low n numbers, we were not able to analyze by gender to rule out these effects in the present experiments.”
- The values and the florescence photos are not match. Figure 7B, the BIF values of 4h and 72h in Fat-CR KO mice are both 2.75, however, from the immunoreactivity photos, red florescence is much higher at 72h compare to 4h. Figure 10B, from the immunoreactivity photos, WT-Norm 0h and FAT-CRKO 0h showed no difference, the value are 0.8 VS 3.1? And the peak value is 12.6 at WT-Norm 4h, however, florescence density isn’t higher than FAT-CRKO at 4h and 72h.
Response: Thank you for your helpful comment. We apologize for not using an image that most accurately represented the averaged values. In the revised version of the manuscript, we have now replaced the image used in Figure 7 at 4 h for the Fat-CR KO and WT-Norm (which also should have similar immunoreactivity with an average score of 2.75). We also changed the microphotographs used for WT-Norm 0 h and Fat-CR KO at 72 h in Figure 10 to more accurately represent the present results. Unfortunately, given low group numbers and high variation, averaged result were sometimes difficult to be represented by microphotographs. This was the case in Figure 10 for WT-Norm 4 h and Fat-CR KO. Figures used for 4h-120h are representative of the 10x field of view. To better represent differences between WT-Norm at 4 h and Fat-CR KO at 4 h and 72 h, we now utilized a different representative image at 72 h for Fat-CR KO and brightened the overall contrast of the green channel by 15% for the whole respective Figure panel to enhance clarity of MPO positive staining for the reader.
Comment: Figure 10B, from the immunoreactivity photos, WT-Norm 0h and FAT-CRKO 0h showed no difference, the value are 0.8 VS 3.1? And the peak value is 12.6 at WT-Norm 4h, however, florescence density isn’t higher than FAT-CRKO at 4h and 72h.
Response: We appreciate your careful considerations of the graphs used in our figures and apologize for not being clear enough when explaining comparisons. AT 0 h WT-Norm and Fat-CR KO did not show any statistical significance following analysis using a two-way ANOVA (p-value = 0.1810).
- Discussion part is too long. Macrophage polarization is not related to this study.
Response: Thank you for your critical review that helped us improve our manuscript. We have edited the discussion in an attempt to shorten it and removed the section on macrophage polarization.
References
[1] Mayer, K., Kiessling, A., Ott, J., Schaefer, M. B., Hecker, M., Henneke, I., Schulz, R., Günther, A., Wang, J., Wu, L., Roth, J., Seeger, W., and Kang, J. X. Acute lung injury is reduced in fat-1 mice endogenously synthesizing n-3 fatty acids. American journal of respiratory and critical care medicine. 2009 179, 6, 474–483. DOI=10.1164/rccm.200807-1064OC.
[2] Pflieger, F. J. 2021. Multimodale Analysen von Lipiden, Lipid-Metaboliten und relevanten Enzymen im Gehirn der Maus unter dem Einfluss von ω-3-Fettsäuren während systemischer Entzündung. Edition Scientifique. VVB Laufersweiler Verlag, Gießen.
[3] Matute-Bello, G., Liles, W. C., Radella, F., Steinberg, K. P., Ruzinski, J. T., Hudson, L. D., and Martin, T. R. Modulation of neutrophil apoptosis by granulocyte colony-stimulating factor and granulocyte/macrophage colony-stimulating factor during the course of acute respiratory distress syndrome. Critical care medicine. 2000 28, 1, 1–7. DOI=10.1097/00003246-200001000-00001.
[4] Sahu, B., Sandhir, R., and Naura, A. S. Two hit induced acute lung injury impairs cognitive function in mice: A potential model to study cross talk between lung and brain. Brain, behavior, and immunity. 2018 73, 633–642. DOI=10.1016/j.bbi.2018.07.013.
[5] Gotts, J. E., Bernard, O., Chun, L., Croze, R. H., Ross, J. T., Nesseler, N., Wu, X., Abbott, J., Fang, X., Calfee, C. S., and Matthay, M. A. Clinically relevant model of pneumococcal pneumonia, ARDS, and nonpulmonary organ dysfunction in mice. American journal of physiology. Lung cellular and molecular physiology. 2019 317, 5, L717-L736. DOI=10.1152/ajplung.00132.2019.
[6] Russell, W. and Burch, R. L. The Principles of Humane Experimental Technique. Medical Journal of Australia. 1960 1, 13, 500. DOI=10.5694/j.1326-5377.1960.tb73127.x.
[7] Chen, H., Bai, C., and Wang, X. The value of the lipopolysaccharide-induced acute lung injury model in respiratory medicine. Expert review of respiratory medicine. 2010 4, 6, 773–783. DOI=10.1586/ers.10.71.
[8] Khadangi, F., Forgues, A.-S., Tremblay-Pitre, S., Dufour-Mailhot, A., Henry, C., Boucher, M., Beaulieu, M.-J., Morissette, M., Fereydoonzad, L., Brunet, D., Robichaud, A., and Bossé, Y. Intranasal versus intratracheal exposure to lipopolysaccharides in a murine model of acute respiratory distress syndrome. Scientific reports. 2021 11, 1, 7777. DOI=10.1038/s41598-021-87462-x.
[9] Tonelli, L. H., Holmes, A., and Postolache, T. T. Intranasal immune challenge induces sex-dependent depressive-like behavior and cytokine expression in the brain. Neuropsychopharmacology : official publication of the American College of Neuropsychopharmacology. 2008 33, 5, 1038–1048. DOI=10.1038/sj.npp.1301488.
[10] Bi, M. H., Ott, J., Fischer, T., Hecker, M., Dietrich, H., Schaefer, M. B., Markart, P., Wang, B. E., Seeger, W., and Mayer, K. Induction of lymphocyte apoptosis in a murine model of acute lung injury--modulation by lipid emulsions. Shock (Augusta, Ga.). 2010 33, 2, 179–188. DOI=10.1097/SHK.0b013e3181ac4b3b.
[11] Hecker, M., Ott, J., Sondermann, C., Schaefer, M., Obert, M., Hecker, A., Morty, R. E., Vadasz, I., Herold, S., Rosengarten, B., Witzenrath, M., Seeger, W., and Mayer, K. Immunomodulation by fish-oil containing lipid emulsions in murine acute respiratory distress syndrome. Critical care (London, England). 2014 18, 2, R85. DOI=10.1186/cc13850.
[12] Schaefer, M. B., Pose, A., Ott, J., Hecker, M., Behnk, A., Schulz, R., Weissmann, N., Günther, A., Seeger, W., and Mayer, K. Peroxisome proliferator-activated receptor-alpha reduces inflammation and vascular leakage in a murine model of acute lung injury. The European respiratory journal. 2008 32, 5, 1344–1353. DOI=10.1183/09031936.00035808.
Reviewer 3 Report
In this research article, " n-3 polyunsaturated fatty acids modulate LPS-induced ARDS and the lung-brain axis of communication in wild type versus Fat-1 mice genetically modified for leukotriene B4 receptor 1 or chemerin receptor 23 knock-out” Jessica Hernandez et. al. experimentally investigated that LPS-induced ARDS increased inflammatory signaling in the periphery and was deceptive in the hypothalamus where low humoral signaling and immune cell trafficking to the brain contributed to immune-to-brain communication. A deficiency in RvE1 receptors as well as enhanced production of n-3 PUFAs was able to sufficiently alter the profile of several inflammatory mediators, modify glial activation, and alter lipid mediators in the brain. In vivo, mice models were used for the proof of concept. Several techniques, for example, immunohistochemistry, fluorescence microscopy, Luminex (for cytokine analysis), RT-PCR, LC-MS/MS, and some other methods were used for this study.
The comments and suggestions for this manuscript are as follows-
1. The introduction and discussion of this manuscript appear as a review article; the author needs to be more comprehensive about the “lung and brain axis” and some text volume (traditional information) can be reduced with proper references.
2. Since this manuscript is focused on lung and brain during ARDS, the author should keep all the lung and brain data (changes in inflammatory mediators/ mRNA expression) in the main manuscript. Other data can be shifted to the supplementary files.
3. Figure 2. Was any difference in body weight observed between WT and fat background mice?
4. Page 7, lines 251-252. The statement “Overall, LPS-induced ARDS also increased markers of inflammation in the brain with a peak at 24 h after stimulation” and lines 260-262 the statement “In particular, at 72 h p.i. n-3 PUFA enrichment (Fat background) reduced all markers of oxidative stress when comparing the Norm and CR KO groups regardless of RvE1 receptor expression “. LPS-TLR4 signaling, MD2, NF-kb activation is the well-studied pathway. Is the increased response at 24 hours because of immune system activation due to septic shock and response reduced at 72 hours due to detoxification of LPS from the body? Why a single dose of LPS, while, when pathogens infect the host their number increases exponentially with time, at least the first 3-4 days?
5. The overall intensive work has been done. I wonder why the author did not use live bacteria instead of a single dose of LPS. The data could be more informative, because the sublethal dose of LPS, after a few hours neutralized by the host response.
6. At 10ug LPS intratracheal administration, what is the ratio of neutrophil recruitment in the lung vs. brain?
7. 4.2 Experimental protocol (page 29, line 1123-1125). The statement, “LPS solution (10 ug Lipopolysaccharide, Escherichia coli O111:B4, in 50 ul normal saline/mouse) was instilled into the lung via a 22 G indwelling venous catheter through the trachea”. Why was the trachea selected for LPS instillation? Why not an intraperitoneal or intranasal site? The author should provide an explanation with suitable reference.
8. The overall experimental design and result interpretation was reasonable.
Minor editing of english language required
Author Response
The comments and suggestions for this manuscript are as follows-
- The introduction and discussion of this manuscript appear as a review article; the author needs to be more comprehensive about the “lung and brain axis” and some text volume (traditional information) can be reduced with proper references.
Response: Thank you for your helpful comment, we have edited the introduction and discussion in an attempt to focus more on the “lung and brain axis” and shorten it.
- Since this manuscript is focused on lung and brain during ARDS, the author should keep all the lung and brain data (changes in inflammatory mediators/ mRNA expression) in the main manuscript. Other data can be shifted to the supplementary files.
Response: Thank you for your critical review, which helped to improve the revised version of our manuscript. We agree with the referee and now increased the focus on the lung and brain axis. As such, we shifted the figure of the Multiplex analysis of the liver to supplementary figures and shortened the paragraph.
- Figure 2. Was any difference in body weight observed between WT and fat background mice?
Response: During a previous study in our own lab comparing WT and fat-1 mice, we assessed the baseline weights and did not determine any significant differences. Moreover, we assessed changes after i.p. LPS-stimulation and did not reveal effects between WT and FAT-1 [2]. In the collaborative present study, body weight and body weight changes were not assessed.
- Page 7, lines 251-252. The statement “Overall, LPS-induced ARDS also increased markers of inflammation in the brain with a peak at 24 h after stimulation” and lines 260-262 the statement “In particular, at 72 h p.i. n-3 PUFA enrichment (Fat background) reduced all markers of oxidative stress when comparing the Norm and CR KO groups regardless of RvE1 receptor expression “. LPS-TLR4 signaling, MD2, NF-kb activation is the well-studied pathway. Is the increased response at 24 hours because of immune system activation due to septic shock and response reduced at 72 hours due to detoxification of LPS from the body? Why a single dose of LPS, while, when pathogens infect the host their number increases exponentially with time, at least the first 3-4 days?
Response: We appreciate your interest in our study and are grateful for your constructive comments. Indeed, we suspect that the increased response at 24 h is due to the immune system activation and a prolonged effect in the brain and due to the low LPS dose detoxification of LPS leads to a reduced response already after 72 h. However, the dose of LPS, which was used causes a uniform, self-limiting, moderate inflammation that does not lead to a septic shock. The single dose of LPS mimics only a mild to moderate lung inflammation since the aim of our study was to show that even a mild to moderate lung inflammation has an effect on the brain. In line with this notion, we did only detect signs for mild humoral inflammatory spillover i.e. very modest increases in inflammatory marker proteins in the liver. While we agree with this referee that a single LPS-injection does not mimic the time course of natural bacterial infection, LPS-injection remains a valuable and standardized model that enabled us to gain new insights into lung-brain communication. In the future, for example, multi-hit experiments or live bacterial infection could be an option to expand on our present findings. In response to the comment by this referee, we now added a new paragraph to increase awareness of this limitation by the reader:
page 29, lines 1133 to 1138: „Moreover, different or more severe models of lung inflammation [3] like a multi-hit model consisting of intratracheal application of hydrochlorid acid and LPS [4] or stimulating pulmonary infection with Streptococcus pneumoniae [5] may be a suitable approaches to expand on the results of our present study with the aim to more precisely mimic the human situation such as the time course and components/subtypes of lung inflammation.“.
- The overall intensive work has been done. I wonder why the author did not use live bacteria instead of a single dose of LPS. The data could be more informative, because the sublethal dose of LPS, after a few hours neutralized by the host response.
Response: Thank you for your comment. Indeed, the use of live bacteria would have been very interesting. However due to the widespread use of LPS as a model of ARDS, the model and results are highly comparable internationally. LPS induces inflammation, which is comparable to the inflammatory processes in humans, especially for the immunological response, neutrophil recruitment, cytokine release and disruption of barrier function [3]. In consideration of comparability, existing expertise and continuity, LPS was used instead of live bacteria in this study. While live bacteria are a very meaningful model, variability between mice are known to be higher than after LPS-injections, which is challenging for descriptive, partially explorative studies such as our present study on lung-brain axis. In the future, a model with live bacteria could be a good follow-up of our present study. As stated above, we addressed this comment in the revised version of our manuscript as follows:
Page 29, lines 1133 to 1138: „Moreover, different or more severe models of lung inflammation [3] like a multi-hit model consisting of intratracheal application of hydrochlorid acid and LPS [4] or stimulating pulmonary infection with Streptococcus pneumoniae [5] may be a suitable approach to expand on the results of our present study with the aim to more precisely mimic the human situation such as the time course and components/subtypes of lung inflammation.“.
- At 10ug LPS intratracheal administration, what is the ratio of neutrophil recruitment in the lung vs. brain?
Response: Unfortunately, we cannot provide such data. We agree that the neutrophil count of the lung would have been very interesting for comparison to the brain. Since the experiment relied on the three R principle (Replacement, Reduction, and Refinement), we collected the tissue from a pre-existing experiment. The rest of the lungs was needed for different experiments and we were not able to count the neutrophils in the bronchoalveolar lavage fluid (BALF) or lung tissue. However, other studies have shown the neutrophil count in the BALF after intratracheal LPS instillation [4]. We aim to address this question in future experiment and added the following lines to our manuscript:
Page 28, line 1117 to 1121: „.Overall, we have to acknowledge the screening character of the present study, which made use of the RRR principle of Russel and Burch [6] to perform our analyses on tissue that was harvested in collaboration from ongoing experiments for a different purpose. To reduce the amount of animals used for scientific purposes, we focused on LPS-stimulated mice. In this manner, parts of the lung were used for different experiments and we were not able to quantify neutrophils in the lung.“
- 4.2 Experimental protocol (page 29, line 1123-1125). The statement, “LPS solution (10 ug Lipopolysaccharide, Escherichia coli O111:B4, in 50 ul normal saline/mouse) was instilled into the lung via a 22 G indwelling venous catheter through the trachea”. Why was the trachea selected for LPS instillation? Why not an intraperitoneal or intranasal site? The author should provide an explanation with suitable reference.
Response: We apologize for not having been clear enough about the rationale for choosing intratracheal LPS instillation as our model. Intravenous and intraperitoneal LPS administration does not lead to lung injury or a lung tissue-specific damage, whereas intranasal or intratracheal LPS administration causes acute pulmonary damage in mice [7]. Previous studies showed that an intratracheal exposure is as effective as an intranasal exposure in a murine model of ARDS induced by LPS [8]. Interestingly, there are already studies showing that intranasal application of LPS may result in the development of a pro-inflammatory status in the olfactory bulb, which could lead to atrophy in the olfactory bulb. Therefore, intranasal application presents a model with a direct influence on the brain [9]. Intratracheal application of LPS more precisely mimics lung inflammation of the lower airways, which was the aim of our present study. In addition, by applying intratracheal instillation of LPS, we have better comparability to previous studies working with ARDS. Indeed, our group has long-standing experience with intratracheal installation as previously published [10–12]. To better explain the rationale for choosing intratracheal LPS-application, we now have adjusted the following paragraph and references to the “Methods” section of the revised version for the manuscript:
Page 29, line 1163 to 1170: “Mice were anesthetized (0,25mg/kg KGW Medetomidin/Domitor, 1mg/ml, Elanco, Bad Homburg, Deutschland; diluted in saline, i.p.) and in total 50 µl (in 10 µl – 20 µl – 20 µl portions) LPS solution (10 µg Lipopolysaccharide, Escherichia coli O111:B4, in 50 µl normal saline/mouse) was instilled into the lung via a 22 G indwelling venous catheter though the trachea as previously described [10–12]. Intratracheal application of LPS mimics lung inflammation of the lower airways. Therefore, we have chosen such application route to investigate lung-brain communication.”
- The overall experimental design and result interpretation was reasonable.
Response: Thank you for your critical review, which helped to improve the revised version of our manuscript.
References
[1] Mayer, K., Kiessling, A., Ott, J., Schaefer, M. B., Hecker, M., Henneke, I., Schulz, R., Günther, A., Wang, J., Wu, L., Roth, J., Seeger, W., and Kang, J. X. Acute lung injury is reduced in fat-1 mice endogenously synthesizing n-3 fatty acids. American journal of respiratory and critical care medicine. 2009 179, 6, 474–483. DOI=10.1164/rccm.200807-1064OC.
[2] Pflieger, F. J. 2021. Multimodale Analysen von Lipiden, Lipid-Metaboliten und relevanten Enzymen im Gehirn der Maus unter dem Einfluss von ω-3-Fettsäuren während systemischer Entzündung. Edition Scientifique. VVB Laufersweiler Verlag, Gießen.
[3] Matute-Bello, G., Liles, W. C., Radella, F., Steinberg, K. P., Ruzinski, J. T., Hudson, L. D., and Martin, T. R. Modulation of neutrophil apoptosis by granulocyte colony-stimulating factor and granulocyte/macrophage colony-stimulating factor during the course of acute respiratory distress syndrome. Critical care medicine. 2000 28, 1, 1–7. DOI=10.1097/00003246-200001000-00001.
[4] Sahu, B., Sandhir, R., and Naura, A. S. Two hit induced acute lung injury impairs cognitive function in mice: A potential model to study cross talk between lung and brain. Brain, behavior, and immunity. 2018 73, 633–642. DOI=10.1016/j.bbi.2018.07.013.
[5] Gotts, J. E., Bernard, O., Chun, L., Croze, R. H., Ross, J. T., Nesseler, N., Wu, X., Abbott, J., Fang, X., Calfee, C. S., and Matthay, M. A. Clinically relevant model of pneumococcal pneumonia, ARDS, and nonpulmonary organ dysfunction in mice. American journal of physiology. Lung cellular and molecular physiology. 2019 317, 5, L717-L736. DOI=10.1152/ajplung.00132.2019.
[6] Russell, W. and Burch, R. L. The Principles of Humane Experimental Technique. Medical Journal of Australia. 1960 1, 13, 500. DOI=10.5694/j.1326-5377.1960.tb73127.x.
[7] Chen, H., Bai, C., and Wang, X. The value of the lipopolysaccharide-induced acute lung injury model in respiratory medicine. Expert review of respiratory medicine. 2010 4, 6, 773–783. DOI=10.1586/ers.10.71.
[8] Khadangi, F., Forgues, A.-S., Tremblay-Pitre, S., Dufour-Mailhot, A., Henry, C., Boucher, M., Beaulieu, M.-J., Morissette, M., Fereydoonzad, L., Brunet, D., Robichaud, A., and Bossé, Y. Intranasal versus intratracheal exposure to lipopolysaccharides in a murine model of acute respiratory distress syndrome. Scientific reports. 2021 11, 1, 7777. DOI=10.1038/s41598-021-87462-x.
[9] Tonelli, L. H., Holmes, A., and Postolache, T. T. Intranasal immune challenge induces sex-dependent depressive-like behavior and cytokine expression in the brain. Neuropsychopharmacology : official publication of the American College of Neuropsychopharmacology. 2008 33, 5, 1038–1048. DOI=10.1038/sj.npp.1301488.
[10] Bi, M. H., Ott, J., Fischer, T., Hecker, M., Dietrich, H., Schaefer, M. B., Markart, P., Wang, B. E., Seeger, W., and Mayer, K. Induction of lymphocyte apoptosis in a murine model of acute lung injury--modulation by lipid emulsions. Shock (Augusta, Ga.). 2010 33, 2, 179–188. DOI=10.1097/SHK.0b013e3181ac4b3b.
[11] Hecker, M., Ott, J., Sondermann, C., Schaefer, M., Obert, M., Hecker, A., Morty, R. E., Vadasz, I., Herold, S., Rosengarten, B., Witzenrath, M., Seeger, W., and Mayer, K. Immunomodulation by fish-oil containing lipid emulsions in murine acute respiratory distress syndrome. Critical care (London, England). 2014 18, 2, R85. DOI=10.1186/cc13850.
[12] Schaefer, M. B., Pose, A., Ott, J., Hecker, M., Behnk, A., Schulz, R., Weissmann, N., Günther, A., Seeger, W., and Mayer, K. Peroxisome proliferator-activated receptor-alpha reduces inflammation and vascular leakage in a murine model of acute lung injury. The European respiratory journal. 2008 32, 5, 1344–1353. DOI=10.1183/09031936.00035808.
Round 2
Reviewer 2 Report
Because of low group numbers, the results showed high variation. The conclusions are not convinced.
Figure 8, CR-KO mice (both WT and FAT mice) showed high neutrophils BIF compared to Normal mice at 0h. Did CR knockout cause high basal level of inflammation even without LPS?
And the result of Figure 8 is not in context. Or line 497 Figure 11 should be Figure 8?
Author Response
Comments and Suggestions for Authors
Because of low group numbers, the results showed high variation. The conclusions are not convinced.
Response: Thank you for your critical review, which helped to improve the revised version of our revised manuscript. In response to this comment, we now increased the focus on the lung and brain axis and the potential for communication in our conclusion. Additionally, we have tried to improve the language we used in our conclusion and refrained from using definite terminology as follows:
page 34, lines 1364 – 1380: “In our hands, even with some of our present results being descriptive, we revealed that LPS-induced ARDS increased inflammatory signaling in the lung and the hypothalamus where low humoral signaling and immune cell trafficking to the brain potentially contributed to immune-to-brain communication (Figure 10). A deficiency in RvE1 receptors as well as enhanced production of n-3 PUFAs was able to sufficiently alter the profile of several inflammatory mediators, modify GFAP immunoreactivity (for CR deficiency) and alter lipid mediators in the brain. In conclusion, we have presented evidence that genetic enrichment of n-3 PUFAs and RvE1 receptor deficiencies can alter the lipid profile in the brain during lung inflammation. In particular, acting through CR, RvE1 may be able to modulate communication to some degree through the humoral pathway and neutrophil recruitment. Moreover, we suggest a role for n-3 PUFAs to modulate neutrophil recruitment to the brain during lung inflammation. Overall, n-3 PUFAs and in particular SPMs i.e. NPD1/PDX or Mar1 and SPM precursors such as 18-HEPE, 17(S)-HDHA or 14(S)-HDHA, respectively, may represent good targets for future investigations to modulate inflammation in individuals suffering from or at risk for developing brain inflammation as a result of ARDS.”
Figure 8, CR-KO mice (both WT and FAT mice) showed high neutrophils BIF compared to Normal mice at 0h. Did CR knockout cause high basal level of inflammation even without LPS?
Response: We appreciate your careful considerations of the graphs used in our figures. Although higher amounts of neutrophils were counted in the BIF of CR-KO mice, we did not detect any other sign of increased inflammatory signaling in lung or hypothalamus in CR-KO mice at basal levels. This is in line with the findings of Bondue et al [1] who also did not show increased inflammation in the lung of CR-KO mice compared to WT mice at baseline. Nonetheless, it is an interesting finding and we aim to further investigate neutrophil recruitment to the brain after LPS instillation to the lung in future studies.
And the result of Figure 8 is not in context. Or line 497 Figure 11 should be Figure 8?
Response: Thank you for pointing out this mistake. We apologize for the typo and have changed the incorrect labelling accordingly; Figure 8 instead of Figure 11 as follows:
Page 13, lines 494-497 “Upon further analysis at the level of the BIF, n-3 PUFAs did not significantly affect neutrophil recruitment over time; however, at 0 h (p = 0.0441) and 72 h p.i. (p = 0.0274) n-3 PUFA enrichment (Fat background) in CR KO mice did show higher levels of neutrophils compared to Fat-Norm controls (Figure 8).”
In general: Upon reviewing our Figures again, we found that the box in the schematic (A) identifying the bifurcation (BIF) for Figure 5 and Figure 8 had shifted so we adjusted the schematic for these two figures for more accuracy. Additionally, in the supplementary Figures we updated Figure S6 to a more accurate representative image for WT-CR KO 0h. Similarly, Figure S11 was updated to better represent WT-Norm 0h, which was already adjusted for Figure 8 in the first revision. Please note that we also added sentences in the legends for supplementary Figures involving immunofluorescence staining acknowledging comparisons made with control animals and CR KO or LR KO (Figures S7, S8, S10 and S11 e.g. “Please note that data on WT-Norm and Fat-Norm controls are displayed again for comparison.”).
References
[1] Bondue, B., Vosters, O., Nadai, P. de, Glineur, S., Henau, O. de, Luangsay, S., van Gool, F., Communi, D., Vuyst, P. de, Desmecht, D., and Parmentier, M. ChemR23 dampens lung inflammation and enhances anti-viral immunity in a mouse model of acute viral pneumonia. PLoS pathogens. 2011 7, 11, e1002358. DOI=10.1371/journal.ppat.1002358.